# UE4-NeRF:Neural Radiance Field for Real-Time Rendering of Large-Scale Scene

**Jiaming Gu**[1,2]* **Minchao Jiang**[1]* **Hongsheng Li**[1] **Xiaoyuan Lu**[2] **Guangming Zhu**[1]
**Syed Afaq Ali Shah**[3] **Liang Zhang**[1]† **Mohammed Bennamoun**[4]
[1]School of Computer Science and Technology, Xidian University
[2]Algorithm R&D Center, Qing Yi (Shanghai)
[3]Edith Cowan University
[4] The University of Western Australia
jiaming_gu_xidian@outlook.com, {jamchaos, hsli}@stu.xidian.edu.cn
xylu@bnc.org.cn, {gmzhu, liangzhang}@xidian.edu.cn
afaq.shah@ecu.edu.au, mohammed.bennamoun@uwa.edu.au

## Abstract

Neural Radiance Field (NeRF) is an implicit 3D reconstruction method that has shown immense potential and has gained significant attention for its ability to reconstruct 3D scenes solely from a set of photographs. However, its real-time rendering capability, especially for interactive real-time rendering of large-scale scenes, has significant limitations. To address this challenge, we propose a novel neural rendering system called UE4-NeRF that is designed for real-time rendering of large-scale scenes. Our proposed approach partitions large scenes into sub-NeRFs, and uses polygonal meshes to represent them. In order to represent the partitioned independent scene, we initialize polygonal meshes by constructing multiple regular octahedra within the scene and the vertices of the polygonal faces are continuously optimized during the training process. Drawing inspiration from the Level of Detail (LOD) techniques, we train meshes with varying levels of detail for different observation levels. Our approach combines with the rasterization pipeline in Unreal Engine 4 (UE4), achieving real-time rendering of large-scale scenes at 4K resolution with a frame rate of up to 43 FPS. Our experimental results demonstrate that our method attains rendering quality on par with state-of-the-art approaches, while additionally offering the advantage of real-time performance. Furthermore, rendering within UE4 facilitates scene editing in subsequent stages. Project page: https://jamchaos.github.io/UE4-NeRF/.

## 1 Introduction

Real-time interactive rendering capability for 3D large-scale scenes is a crucial tool in creating digital worlds for applications such as virtual reality (VR), computer games, and the Metaverse. Neural Radiance Fields (NeRF) [23] is a pioneering method for 3D reconstruction and synthesis of novel views, which is different from traditional 3D reconstruction techniques that represent scenes using explicit expressions such as point clouds, grids, and voxels [5, 4, 14, 1]. NeRF samples each ray and retrieves the 3D location of each sampling point and the 2D viewing direction of the ray. Subsequently, these 5D vector values are fed into a neural network to determine the color and volume density of each sampling point. NeRF constructs a field parameterized by an MLP neural network [16, 34] to continuously optimize parameters and reconstruct the scene. NeRF has demonstrated remarkable

---

*Contribute equally

†Corresponding author

37th Conference on Neural Information Processing Systems (NeurIPS 2023).

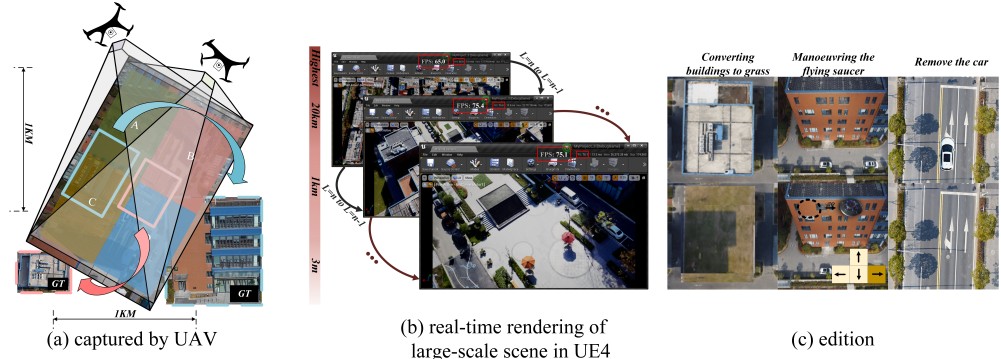

(a) captured by UAV      (b) real-time rendering of      (c) edition
large-scale scene in UE4

Figure 1: (a) Large-scale scenes characterized by complex textures and significant height differences are captured with monocular or multi-camera drone footage. We divide scene into multiple individual sub-NeRFs.(b) Real-time interactive rendering in UE4 with distinct levels of detail, along with frame rate disparities among them. (c) Defferent editing capabilities on the scenes.

performance for 3D reconstruction when provided with a set of posed camera images [29, 2, 23], but early NeRF works [2, 23] had only focused on small-scale scenes and object-centric reconstruction due to the model computational complexity. Although Block-NeRF [37] and BungeeNeRF [43] are capable of reconstructing large-scale environments, they are still unable to achieve real-time rendering. Traditional NeRF obtains the color of each pixel through volume rendering, but this rendering process is far too slow for interactive visualization and compromises high fidelity. Recently, Mobile-NeRF [6] combined with classic polygon rasterization pipeline has shown to achieve real-time rendering on various devices. However, training just a small-scale scene incurs significant overheads, such as training time and GPU resources. It requires 8 v100 GPUs to train the model successfully over several days. Additionally, due to opacity binarization, Mobile-NeRF cannot handle outdoor scenes with a variety of materials, including transparent or semi-transparent objects such as water, glass and plastic shed. Performing subsequent editing on the scenes generated by NeRF is another meaningful but challenging task, and previous works [13, 44, 19] required training complex neural networks to achieve this capability. These methods are time-consuming and are limited in their editing capabilities.

This paper introduces UE4-NeRF to address the aforementioned issues regarding large-scale scene rendering and editing. UE4-NeRF combines NeRF with the Unreal Engine 4 (UE4) [17, 30, 17] to achieve real-time interactive rendering of large-scale scenes with the scene editing ability. As shown in Figure 1(a) , to achieve this, large-scale scene such as industrial parks with large height differences, roads and farmlands are partitioned into separately trained blocks. The size of individual scenes captured by drones is in the range of several square kilometers. The multi-resolution hash encoding proposed in Instant-NGP [25] is employed for training acceleration. In order to efficiently utilize the parallel computing capabilities of GPUs, the computationally intensive portions are implemented using CUDA. Through these efforts, the training speed of UE4-NeRF is $1000\times$ faster than Mobile-NeRF. For each block, we represent the scene using textured polygonal meshes obtained from regular octahedrons. Similar to Mobile-NeRF, the texture atlas stores feature vectors and opacity information. We utilize an MLP as the High-Level Shading Language(HLSL) fragment shader in UE4, which takes in the feature vectors and outputs colors. Furthermore, to ensure real-time rendering performance for different observation heights viewpoints, we adopted a method akin to the Level of Detail (LOD) [8, 21] approach commonly used in computer graphics, where the complexity of 3D model representation is quantified in terms of metrics such as geometric detail and texture resolution. Figure 1(b)  demonstrates that UE4-NeRF is capable of achieving real-time rendering at different relative observation heights by training different levels of polygon meshes, where coarser meshes are used for relatively higher observation heights. One of the other advantages of combining NeRF with UE4 is the powerful development capabilities of the rendering engine, allowing further processing of the final scene rendered in UE4. Users can freely combine different sub-NeRFs and add objects to the scene as needed. Moreover, object manipulation, as illustrated in Figure 1(c) , is also supported.

In summary, our paper presents the following contributions:(1) We propose a novel UE4-NeRF technique which has 1000 times faster training times than MobileNeRF, while maintaining higher

quality and smaller storage overhead;(2) The demonstration of the capability of UE4-NeRF to reconstruct large-scale scenes in real-time and showcase the practical viability of NeRF;(3) The integration of NeRF and UE4 facilitates editing and manipulation of 3D scenes.

## 2 Related work

**Large Scale 3D Reconstruction.** The original NeRF algorithm was limited to object-centric scenes or scenes with boundaries. NeRF++ [46] expanded upon NeRF by introducing an inverted sphere parameterization to handle unbounded scenarios. Mip-NeRF 360 [3] addressed the challenges associated with unbounded scenes through the use of non-linear scene parameterization, online distillation and a novel distortion-based regularizer. However, it was observed that simply expanding the scene without adjusting the model capacity, i.e., using a fixed-size MLP, can lead to unstable training and loss of high-fidelity. Meanwhile, naively increasing the model capacity can significantly increase the training cost. BungeeNeRF [43] proposed a progressive approach to build and train large-scale scene models, which promoted layer specialization and unleashed the power of positional encoding over the full frequency range. Block-NeRF [37] and Mega-NeRF [40] spatially partitioned a large scene into several individually trained sub-NeRFs. They also incorporated appearance embeddings like NeRF-W [22] to handle appearance variations in the scene. Similar to Block-NeRF, our approach partitions the large-scale scene into sub-NeRFs, and trains a coarse scene model to assist in accurately delineating each sub-NeRF. *However, our approach is not only applicable to the grid structure like Block-NeRF, but is also generally suitable for various types of scenes, e.g. mountainous terrain. Additionally, in contrast to the aforementioned NeRFs designed for large-scale scenes, our method accomplishes real-time rendering.*

**Rendering of Neural Fields.** NeRF has tremendous potential for rendering and has gained attention for 3D reconstruction from a set of posed camera images [29, 2, 23]. However, the computational overhead of NeRF limits its practical applications. Some prior works [25, 11] have addressed training costs, while others have focused on rendering [9, 36, 27, 31]. SNeRG [20] and PlenOctrees [45] improved performance by baking the components of NeRF. However, these methods do not take sufficient advantage of the texture-based rasterization pipeline and require significant GPU memory to store volumetric textures. MobileNeRF [6] is a variation of NeRF that can run in real-time on a variety of common mobile devices. To render an image, Mobile-NeRF utilized the classic polygon rasterization pipeline with Z-buffering to produce a feature vector for each pixel and then passed it to a lightweight MLP running in a GLSL fragment shader to produce the output color. *In this work, we accelerate the rendering of NeRF by converting the learned knowledge into a mesh representation for efficient mesh rasterization pipelines. Moreover, we achieve a significant milestone by successfully implementing NeRF for real-time interactive rendering of large-scale scenes, which was previously unexplored.*

**Manipulation and Composition of NeRF.** Practical applications of 3D representations require effective manipulation and composition capabilities. While, explicit 3D representations, such as meshes and voxels, inherently support editing and composition, it becomes challenging for neural network-based implicit representations such as vanilla NeRF [23]. NSVF [20] uses a sparse voxel octree for modeling, employing voxel embeddings for each scene while sharing the same MLP for predicting density and color. However, the flexibility of this approach is somewhat constrained. GIRAFFE [28], combined with GAN [49, 18], enabled the addition of multiple objects in a scene, expanding from single-object to multi-object generation, even in the absence of such objects in the training data. Plenoxels [11] utlizes explicit sparse voxel representation for direct object composition of different objects, but suffers from large storage requirements of the dense index matrix. None of these methods support object manipulation in a 3D scene. *In contrast, our approach leverages UE4 to achieve large-scale scene rendering, enabling users to freely add and manipulate objects within the UE4-NeRF environment. For instance, users can use a broomstick to fly or drive a car within the rendered environment powered by UE4-NeRF.*

## 3 Proposed Method

Our novel method involves representing large scenes as polygon meshes, where the positions of mesh vertices are iteratively optimized during the training phase. The encoding network generates texture

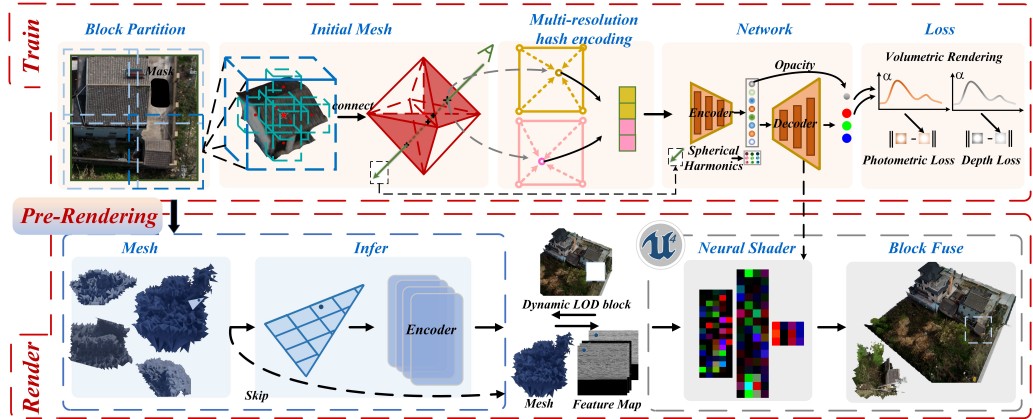

Figure 2: **Overview of UE4-NeRF.** UE4-NeRF consists of three modules: (1) In the training module, we partition the sub-NeRFs and initialize the grids in each partitioned scene. During the training process, we continuously optimize the parameters of the Encoder-Decoder network and the vertex positions of the meshes; (2) Through the pre-rendering module, we extract the polygonal meshes at different levels of detail that will be used for the final rendering; (3) The rendering module consists of an inference submodule and an UE4 submodule, which communicate with each other to achieve the final real-time rendering.

maps that store opacity and feature vectors, essential for rendering. To accomplish real-time rendering of large scenes, we combine the pre-rendered polygon meshes with the rasterization pipeline available in Unreal Engine 4 (UE4). In the subsequent subsections, we present a detailed and comprehensive description of entire procedure.

## 3.1 Training

### 3.1.1 Block Partition

As depicted in Figure 2, a large scene is partitioned into multiple blocks. To ensure smooth transitions at the boundaries of the model, each training area, indicated by dashed lines, is slightly larger than its corresponding scene area, with overlapping borders with neighboring blocks. We train a coarse model for sub-region segmentation, focusing on capturing the overall terrain and building outlines. We then filter the images associated with the sub-regions and use a mask to exclude pixels outside the scene area.

### 3.1.2 Optimization

We begin by initializing a $128 \times 128 \times 128$ grid, and selecting the center point of each grid along with its six neighboring grids (front, back, left, right, top, and bottom) to create polygonal meshes. ***To address the slow and unstable convergence issue encountered by Mobile-NeRF when dealing with tilted surfaces, we utilize a regular octahedron with 20 faces, including the 8 exterior faces and 12 interior faces.*** This is illustrated in the "Initial Mesh" section of Figure 2. The ray emitted from the camera origin to the pixel is defined as $\mathbf{r}(t) = \mathbf{o} + t\mathbf{d}$ and $\mathbf{d} = (\theta, \phi)$ is the viewing direction of the ray. For each ray emitted from the camera origin to the pixel, we calculate its intersection with the polygonal meshes. The point of intersection serves as the sampling point for that ray. For each block, we define an Encoder-Decoder network:

$$\mathcal{E}(hash(\mathbf{p_i}); \theta_\mathcal{E}) \rightarrow \mathbf{M_i}, \alpha_i \tag{1}$$

$$\mathcal{D}(\mathbf{M_i}, \mathcal{SH}(\mathbf{d_i}); \theta_\mathcal{D}) \rightarrow \mathbf{c_i} \tag{2}$$

The encoder network, denoted as MLP $\mathcal{E}$, takes positional information($\mathbf{p_i}$) encoded by multi-resolution hash functions[25] as input. It generates an 8D feature vector $\mathbf{M_i}$, which incorporates opacity information. On the other hand, the decoder network $\mathcal{D}$ utilizes $\mathbf{M_i}$ and the ray direction as inputs and predicts the color of the sampling point. The direction of the ray is encoded using Spherical Harmonics.

We use the traditional NeRF volume rendering method (Eq. 3) to obtain the predicted color $\hat{\mathbf{C}}(\mathbf{r})$. Due to the complexity of calculating the distance between sampling points, we simplify the process by fixing the distance value. The opacity $\alpha$ is then directly predicted instead of the volume density.

$$\hat{\mathbf{C}}(\mathbf{r}) = \sum_{i=1}^{N} T_i \alpha_i \mathbf{c_i}, \quad T_i = \prod_{j=0}^{i-1}(1 - \alpha_j) \tag{3}$$

Unlike the traditional photometric loss, our proposed photometric loss consists of two components. The first component referred to as $level$-1 photometric loss is calculated using Eq. 4. It quantifies the mean squared error between the predicted colors and the ground truth colors of the corresponding pixel.

$$\mathcal{L}_{rgb}^{part1}(\theta, V_p) = \sum_{r \in \mathcal{R}} \left\| \hat{\mathbf{C}}(\mathbf{r}) - \mathbf{C}(\mathbf{r}) \right\|_2^2 \tag{4}$$

where $V_p$ is the position of the vertices of the polygonal mesh under the corresponding level and $\mathcal{R}$ is the set of rays in each batch. $\mathbf{C}(\mathbf{r})$ represents the ground truth RGB colors for ray $r$. In addition, to improve the proximity of the triangle meshes to the object surface, we design a second component of the photometric loss, denoted as $\mathcal{L}_{rgb}^{part2}$. As depicted in Figure 3, in contrast to the previous sampling rule, we only select sampling points with opacity greater than a threshold $f$. Furthermore, the volumetric rendering process is halted when the accumulated opacity exceeds 0.8, indicating that the remaining opacity $T'$ is less than 0.2. In the initial 10,000 epochs, the threshold $f$ is maintained at 0, and afterward, as shown in Eq. 5, the $f$ increases as the number of epochs increases, but does not exceed 0.3.

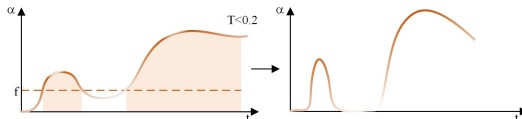

Figure 3: **Illustration of sampling details.** Our approach involves selectively sampling intersection points that are located above a threshold value $f$ and terminating the volume rendering process when the remaining opacity drops below 0.2. This strategy effectively concentrates opacity on a limited number of specific meshes and compresses it towards the object, leading to improved rendering results.

$$f = \min\{0.3, \lfloor \frac{epoch - 10000}{10000} \rfloor^2 \times 0.012\} \tag{5}$$

The photometric loss for the second part of the $level$-1 can be obtained using the following formula:

$$\mathcal{L}_{rgb}^{part2}(\theta, V_p) = \sum_{r \in \mathcal{R}} \left\| \hat{\mathbf{C}}'(\mathbf{r}) - \mathbf{C}(\mathbf{r}) \right\|_2^2, \quad \hat{\mathbf{C}}'(\mathbf{r}) = \frac{\sum_{\phi=1}^{N} T'_\phi \alpha_\phi \mathbf{c}_\phi}{1 - T'_\phi} \tag{6}$$

Where $\hat{\mathbf{C}}'(\mathbf{r})$ is obtained through volume rendering under the aforementioned sampling scheme. To cover the entire color range, we divide the expression by $1 - T'_\phi$ as shown in Eq. 6, since our volume rendering stops when the remaining opacity is less than 0.2. We obtain the final photometric loss of the $level$-1 as follows:

$$\beta \mathcal{L}_{rgb}^{part1}(\theta, V_p) + (1 - \beta) \mathcal{L}_{rgb}^{part2}(\theta, V_p), \quad \beta = 0.8^{\lfloor \frac{epoch}{10000} \rfloor} \tag{7}$$

During the initial 10,000 epochs, the second component of the photometric loss does not have any impact. This is done to align with the traditional volume rendering approach and adhere to the modeling principles of NeRF. ***However, as the training progresses, the weight for the second component gradually increases. This enables the preservation of opacity only on specific meshes and ensures consistency with the later stages, where only meshes with opacity greater than 0.3 are exported for rasterization.***

After 10000 epochs, a process is initiated where we randomly select sampling points within each grid, calculate the opacity of each grid, and select the vertex of the grid with the highest opacity among the eight grids as the vertex of the merged grid. This allows us to generate grids of 64×64×64, 32×32×32, 16×16×16, and 8×8×8 sequentially. Each grid level is tailored for specific observation heights, enabling us to minimize the rendering cost of the scene. It is important to note that for the same block, different levels always use the same network, but the gradient in the rough levels are

only used to update mesh vertex coordinates. This measure can prevent coarse-scale meshes from providing undue supervision to shared networks when they cannot represent fine object structures. The final loss function of the photometric part is calculated as follows:

$$\mathcal{L}_{rgb} = \sum_{l=1}^{5} \beta \mathcal{L}_{rgb}^{part1}(\theta, V_p) + (1 - \beta)\mathcal{L}_{rgb}^{part2}(\theta, V_p) \tag{8}$$

**Pseudo-depth.** Utilizing depth information [10, 33] as a supervisory signal has demonstrated the ability to enhance the convergence rate and improve the quality of reconstructions. However, when operating under the influence of ambient light in outdoor environments, RGB-D cameras are susceptible to produce inaccurate depth measurements and have limited measurement ranges. In our approach, we propose to use the pseudo-depth as the supervision information for NeRF. In conventional NeRF, structure-from-motion (SFM) [35] solvers such as COLMAP [15, 39, 12] are typically utilized to estimate the camera poses. As part of this process, a sparse point cloud can be obtained. By aligning the camera and point cloud in the same coordinate system, we calculate the pseudo-depth by measuring the distance between the sparse point cloud and the camera position. For a detailed explanation of the pseudo-depth loss ($\mathcal{L}_D$) calculation, please refer to the supplementary material. The final loss $\mathcal{L}$ of the model is calculated as follows:

$$\mathcal{L} = \mathcal{L}_{rgb} + \mathcal{L}_D \tag{9}$$

**Transient Objects.** Moving/dynamic objects, such as pedestrians and cars, have the potential to cause inconsistency in the observed scene across different viewpoints and can also create gaps or "holes" in the depth map. To address this challenge, we integrate a semantic segmentation model [7, 48] into our training process. By leveraging this model, we generate masks that identify the regions occupied by the moving objects, effectively excluding them from the training of the scene. This strategy mitigates the impact of moving objects on the scene's depth map, and thus improves the overall fidelity and generalization performance of the model.

## 3.2 Pre-rendering

To optimize computational efficiency, we leverage the predicted values from the acceleration grid to identify and remove meshes that do not possess significant geometric surfaces. Once the training phase is complete, we obtain the network weights as well as the meshes with optimized vertex positions. Subsequently, we perform the pre-rendering process. The overall workflow is illustrated in Figure 4. ***First***, for each block, in addition to the camera view used for training, we incorporate parallel lights from various angles above. We compute the intersection of each ray with the polygon mesh, and the traversal of light is halted when the cumulative

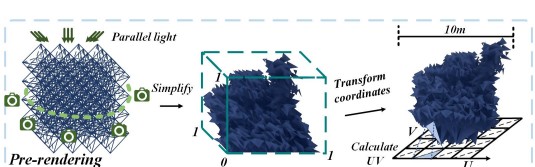

Figure 4: **Structure of the pre-rendering.** We employ a multi-viewpoint approach to extract polygonal faces, generating multiple rays from different perspectives. Subsequently, we apply scaling transformations and calculate UV coordinates to further process the extracted data.

opacity surpasses 0.8. A triangular face will be clipped if the opacity of all intersection points on that face is below 0.3, resulting in the retention of only those meshes that meet this criterion. From this, we obtain the final polygon mesh representations of the scene. Based on our statistical analysis, the final extracted meshes account for only 5% of the original mesh count. This significant reduction effectively improves mesh utilization, inference speed during rendering, and reduces storage costs. ***Second***, we perform coordinate transformation to ensure the obtained mesh aligns with real-world scale. ***Third***, we map the vertex coordinates of the triangles to their corresponding positions in UV coordinates.

## 3.3 Rendering

After training, we obtain the weights of the Encoder-Decoder network, and refined polygonal meshes at different levels through vertex optimization for each individual block. Subsequently, we perform pre-rendering and extract the final meshes. For each polygonal mesh, we perform rasterization using a

Table 1: **Details of the real-world scene captured by UAV.** The memory requirement increases with the size of the scene, while the variation in FPS is primarily influenced by the materials present in the scene itself.

| | Memory(4k) | | Area($m^2$) | FPS | | CA |
| | Host | GPU | | 2k | 4k | |
|---|---|---|---|---|---|---|
| FL | $\sim 12$GB | $\sim 5$GB | 180×240 | 55 | 36 | 120$m$ |
| CS | $\sim 25$GB | $\sim 11$GB | 420×240 | 66 | 43 | 70$m$ |
| IP | $\sim 40$GB | $\sim 14$GB | 600×600 | 58 | 35 | 120$m$ |

Table 2: **Quantitative Comparisons with Existing Methods.** Our method achieves remarkable results in terms of both rendering quality and speed. Among the compared methods, UE4-NeRF requires the least GPU memory. The training costs are also provided.

| Method | PSNR↑ | SSIM↑ | LPIPS↓ | FPS(2K) | Memory(GB) Host | GPU | time(mins) |
|---|---|---|---|---|---|---|---|
| NeRFacto [38] | 20.99 | 0.663 | 0.389 | 0.5 | $\sim 14$ | $\sim 3$ | $\sim 45$ |
| Instant-NGP [25] | 22.00 | 0.631 | 0.426 | 11 | $\sim 10$ | $\sim 32$ | $\sim 15$ |
| Mega-NeRF [40] | 17.37 | 0.23 | 0.546 | ✗ | - | - | $\sim 2160$ |
| Mobile-NeRF [6] | 16.92 | 0.23 | 0.419 | ✗ | - | - | $\sim 2880$ |
| UE4-NeRF | **25.03** | **0.704** | **0.287** | **55** | $\sim 19$ | $\sim$ **3** | $\sim 40$ |

texture map of size 32×32. However, storing texture maps for numerous polygonal meshes in a large-scale scene can lead to significant storage overhead. To address this, we utilize dynamic inference instead of directly storing all texture maps, as done in Mobile-NeRF [6]. Real-time rendering is achieved through continuous and efficient communication between the inference submodule and the UE4 submodule. During the block composition process, since the large-scale scene is accurately divided into multiple blocks and scene fusion is also considered during training, transitions between different blocks in a large-scale scene are smooth and interpolation is not required at the junction.

**Texture map.** Prior to rasterizing the faces of the polygonal mesh, we perform sampling on each mesh and use the encoder network to obtain 8 channels feature vectors. These 8 channels along with the opacity $\alpha$ obtained are processed to generate nine single-channel texture maps using BC4 compression [26]. Compared to the original texture map, this reduces 2/3 of the GPU memory consumption during rendering without significantly compromising the image quality. *Please note that texture maps do not store colors, but rather spatial features, which also contain transparency information.* Taking into account the principle of locality, we employ a strategy where we collectively infer and store information from nearby regions into texture maps. This enables us to directly access the texture maps without the need for re-inference when traversing these regions again.

**Rasterization.** We utilize the traditional rasterization pipeline in UE4 to rasterize each face of the polygonal mesh. Subsequently, the decoder network is applied to convert the 17D features of each pixel into RGB colors (see UE4 submodule in Figure 2). The 17D features comprise 8 channels of learned features and a 9D view direction, which is encoded using spherical harmonic functions [11, 24]. The decoder network is implemented as a HLSL fragment shader of the UE4, which is called Neural Shader. To achieve translucent effects without the need for volume rendering, we employ alpha-dithered [42] and temporal anti-aliasing, which offers a relatively low-overhead approach.

**Dynamic inference.** The interaction between the UE4 submodule and the inference submodule is depicated in Figure 2. When a block requiring rendering is encountered, the UE4 submodule supplies the block and level index information, while the inference submodule provides the corresponding regional meshes and feature maps. In cases where feature maps are not readibly for a specific region, the inference operation is carried out to generate them. This communication mechanism ensures efficient and prompt processing between these two submodules.

## 4 Experiments

### 4.1 Implementation Details

Similar to the Instant-NGP approach, we employ multi-resolution hash encoding and construct an Encoder network comprising a 4-layer MLP (32×64,64×64,64×64,64×8) to estimate the opacity and an 8-dimensional feature vector. Additionally, we develop a Decoder network with 3-layer MLP (17×16,16×16,16×3) to predict the final color of the sampled points. For each block, we train the model using one Nvidia RTX 3090 GPU for 80,000 epochs to achieve convergence, with an approximate duration of 40 minutes.

**Datasets and Metric.** Our objective is to enable real-time rendering of large-scale scenes with complex textures and materials. Table 1 presents three distinct scenarios, namely Farmland(FL),

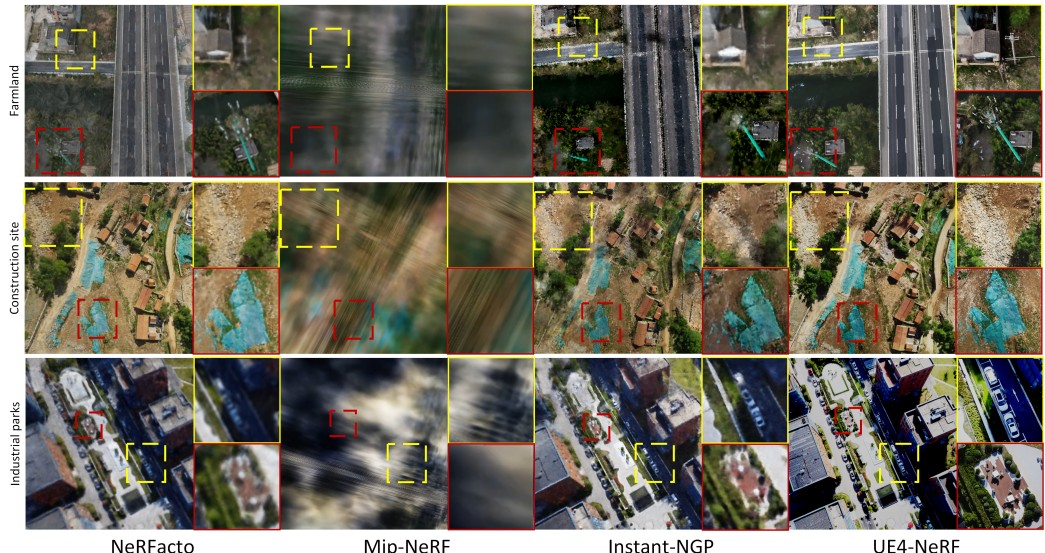

Figure 5: **Qualitative Comparisons with Existing Methods.** We visually compare our proposed technique with recently implemented real-time rendering NeRF models [25, 38] and Mip-NeRF [3]. All these methods are implemented within nerfstudio [38]. It is evident that UE4-NeRF yields the best reconstruction for all the three scenes.

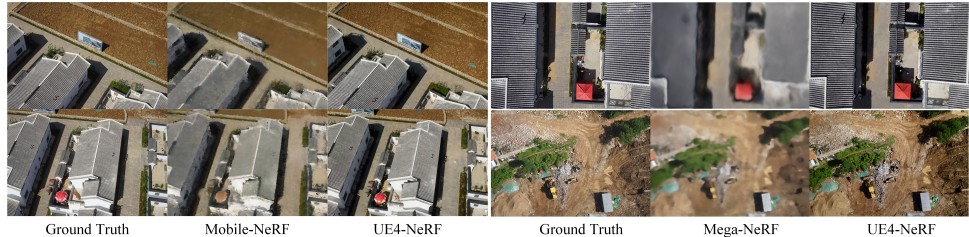

Figure 6: **Qualitative Comparisons with Mega-NeRF and Mobile-NeRF.** Detailed comparison with Mobile-NeRF and Mega-NeRF. Since these methods are very time-consuming, we compared them in a block or scene.

Construction site(CS) and Industrial parks(IP) which were captured using drones to address the challenges that UE4-NeRF was designed to tackle. Each captured image has an approximate resolution of 6000×4000 pixels and contains GPS information. One of our goals is to establish a measurable NeRF model for these scenarios, which requires GPS data for real-world scale conversion. We performed a coordinate system transformation prior to modeling. In the UE4 coordinate system, the positive direction of the X-axis points to the east, the positive direction of the Y-axis points to the south, and the positive direction of the Z-axis points up. The scale is 100:1. We also performed comparative experiments on the dataset introduced in Mega-NeRF[40]. At $level$-1, UE4-NeRF can achieve an maximum frame rate of around 43 FPS at 4K resolution. Due to the limited battery life of the drone, the captured areas were constrained in size. Nevertheless, subsequent experiments have revealed that limitation does not represent the upper limit of UE4-NeRF. In fact, UE4-NeRF can render even larger-scale scenes in real-time. To evaluate the reconstruction quality and fidelity of our model, we use evaluation metrics such as PSNR/SSIM [41] and LPIPS [47]. We also compare the real-time rendering speed of UE4-NeRF with other models when rendering vast scenes.

## 4.2 Comparisons with Existing Methods

We performed qualitative comparisons between UE4-NeRF and other NeRF implementations on four scenes captured by UAV, and the results are depicted in Figure 5. Existing tools like NeR-Facto [38] and Instant-NGP [25] aim to achieve real-time rendering, but tend to procude subpar

Table 3: **Quantitative Comparison with Mega-NeRF and Mobile-NeRF.** Mobile-NeRF can not support complete scene. The maximal resolution supported by Mega-NeRF is 800*800 on RTX 3090.

| Methods | Training time | | FPS(1×3090) |
| --- | --- | --- | --- |
| | block | scene(n blocks,4×3090) | |
| Mobile-NeRF | 48h(4×3090) | n×48h | 35(block,2K) |
| Mega-NeRF | 36h(1×3090) | 12h+36h×n/4 | 60(scene,800×800) |
| UE4-NeRF | **40mins(1×3090)** | **1h+40mins×n/4** | **50(scene,2k), ≥70(block,2K)** |

results when applied to large-scale scenes, as evident in Figure 5. Mip-NeRF, despite increasing the number of training iterations, still exhibits noticeable blurriness when rendering natural scenes. In contrast, UE4-NeRF showcases remarkable accuracy in rendering small objects, such as power lines, in the "farmland" scene. It also excels in handling dense textures, as demonstrated in the "construction site" scene. The "construction site" scene also showcases the rendering effects of translucent objects. In Table 2, we provide a quantitative comparison of our proposed technique with other existing methods using different metrics. It is important to note that the actual rendering quality of UE4-NeRF is expected to surpass the metric's performance,

as achieving consistent exposure matching with the original image is challenging within the Unreal Engine environment. Despite this limitation, UE4-NeRF still outperforms other methods by a significant margin and its real-time rendering capability surpasses that of the current state-of-the-art real-time rendering models. In Table 3, we see that Mobile-NeRF takes 2 days to train just one block and requires 4x3090ti GPUs. If it trains the whole scene, it takes two months. For each block, Mega-NeRF needs 36 hours to train for 500,000 epochs, while we only need 40 minutes to train for 80,000 epochs to reach convergence. When training the entire scene, Mega-NeRF requires 12 hours of additional time, but UE4-NeRF only needs 1 hour. It is worth noting that Mega-NeRF generates hundreds of GBs of temporary files during training. And as can be seen from Figure 6 and Figure 7, compared with Ground truth, UE4-NeRF shows amazing rendering results, but Mobile-NeRF and Mega-NeRF produce significant blur.

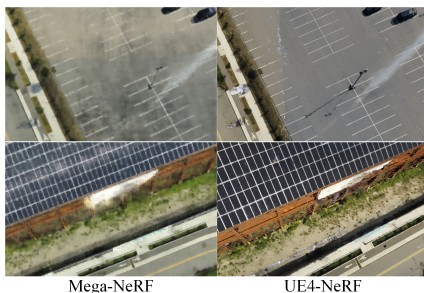

Mega-NeRF                    UE4-NeRF

Figure 7: **Comparison with Mega-NeRF on the public dataset.** Due to the large exposure gap between images in the Mega-NeRF dataset, different shadows appear at different viewing angles.

### 4.3 LOD render results

Figure 8 showcases the qualitative rendering results at different levels of detail, along with the corresponding real-time frame rates, at a resolution of 2K. As the level of hierarchy decreases, the rendering quality and clarity progressively improve. UE4-NeRF demonstrates a high level of fidelity when compared to the ground truth. In addition, it can be observed that the LOD approach effectively

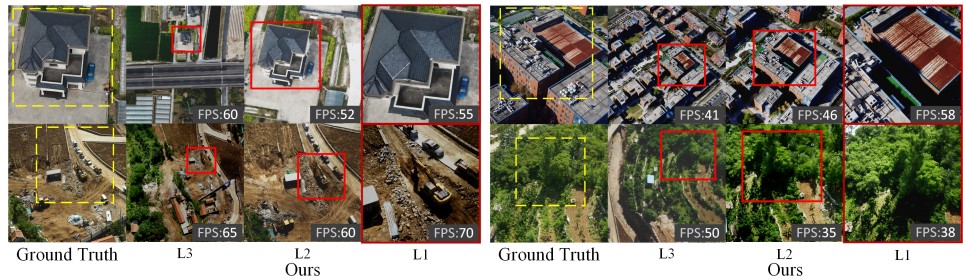

Figure 8: **Rendering result at different LOD.** The red boxes indicate the presentation of the same content at different levels. In addition to achieving high FPS rendering at different LOD in 2K resolution, UE4-NeRF also demonstrates a remarkable photo-realistic.

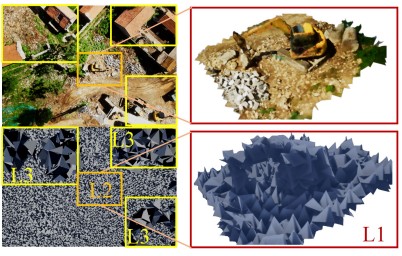
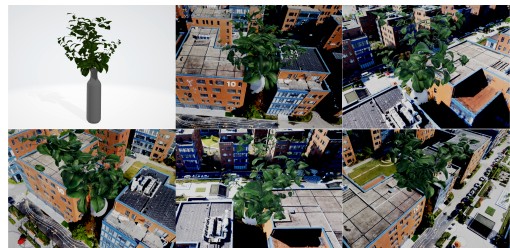

Figure 9: **Shading mesh**– not textured. Polygonal meshes at different levels are displayed, revealing the varying complexity of the scene.

Figure 10: **Scene edition.**The results of multi-view scene editing are presented. One can carefully observe the occlusion relationship between objects and the scene.

addresses the challenge of real-time rendering at higher levels although it may sacrifice some rendering quality at higher levels for the sake of real-time performance. When comparing materials and frame rates across different scenes, it is worth noting that the presence of semi-transparent objects, such as vegetation(as seen in the last scene of Figure 8), can potentially lead to a decrease in frame rate due to increased computational overhead.

**Shading mesh.** Figure 9 showcases the shading meshes extracted without textures. Despite the coarse appearance of the meshes and the limited level of detail even in the closest $level$-1, such as excavators, the final rendering quality remains impressive. This can be attributed to the vertex optimization carried out during the training process, the texture maps encoded in the network outputs, and the deployment of the final neural shader during rendering.

### 4.4   Scene edition

Editing scenes in UE4-NeRF, especially the composition of different scenes, is a seamless process. UE4 supports multiple 3D model file formats, allowing us to import and edit the rendered scenes. The Unreal Engine automatically handles occlusion between the original scene and added objects, making the editing process effortless. We have complete freedom to manipulate the newly added objects according to our needs. In Figure 10, when an object obstructs the NeRF-rendered scene, the occluded parts do not require shading by the neural shader. Interestingly, we observed that after adding and manipulating objects, the frame rate of the scene actually slightly improves. This is attributed to the reduction in NeRF computation, resulting in improved frame rates.

## 5   Conclusion and Limitations

In this research paper, we introduce UE4-NeRF, a real-time interactive rendering system designed for large-scale scenes. UE4-NeRF partitions the scene into blocks and trains a separate NeRF model for each block. Inspired by LOD techniques, different levels of mesh precision are used for different layers, and the NeRF model is integrated with the rasterization pipeline in UE4. Currently, UE4-NeRF stands as the only NeRF model capable of achieving real-time interactive rendering of large-scale realistic scenes while maintaining high fidelity. Moreover, its integration with UE4 opens up possibilities of scene editing and manipulation, allowing for the flexible addition and manipulation of traditional 3D models such as $.obj$ and $.fbx$ within the scene. Overall, UE4-NeRF presents a comprehensive neural rendering system that combines large-scale, real-time interactive rendering with photo-realism and editability. ***However***, there are still some issues that we aim to address in the future: 1) UE4-NeRF requires the direct use of CUDA and therefore relies on NVIDIA GPUs. Rendering large scenes, such as a few square kilometers, incurs significant memory overhead when without reducing rendering accuracy, and increasing rendering accuracy further amplifies memory consumption. 2) During the pre-rendering process, it is difficult to capture rays from any viewing angles and extract meshes for each viewpoint, resulting in gaps or holes in the final rendered scene. The strategy of pre-rendering can be optimized.

## Acknowledgments and Disclosure of Funding

This work is supported by National Natural Science Foundation of China (No.62072358,62073252). The authors would like to express gratitude to the anonymous reviewers who greatly help improve the manuscript. Besides, we sincerely thank Hongli Wang, Ke Zhang and Mengqi Shen for their technical support in drone shooting.

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
