# Supplementary Material

## UE4-NeRF:Neural Radiance Field for Real-Time
## Rendering of Large-Scale Scene

In order to further elaborate our proposed method, this supplementary file provides details about the gradient and pseudo-depth computations, rendering of transparent objects, mesh compression, to name a few. The visualizations of experimental results demonstrate the significance of our proposed method.

## A    Large-Scale Scenes

In Figure 1, we showcase a top-down perspective of large-scale scenes rendered in real-time using UE4-NeRF. The size and contours of the five scenes, with four of them mentioned in the main paper and the fifth being the most recently captured, can be visually perceived in a straightforward manner.

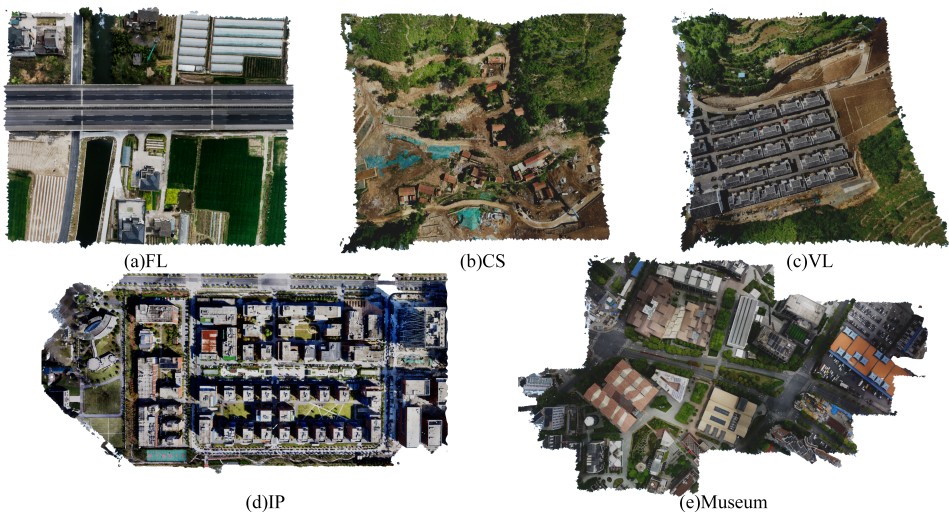

(a)FL          (b)CS          (c)VL

(d)IP          (e)Museum

Figure 1: **Five large-scale scenes rendered in real-time using UE4-NeRF.** We captured four scenes with different textures and complex terrains using a drone, with three of them mentioned in the paper. Each scene can be rendered in real-time using UE4-NeRF.

## B    Qualitative Comparison between MVS and proposed UE4-NeRF

In Figure 2, we have provided additional qualitative comparisons with MVS. Our experiments offer comparisons in transparent objects as well as subtle details. MVS utilizes sparse reconstruction to extract feature points, which are then expanded based on morphological and color differences to generate a dense point cloud. This dense point cloud is further used for surface reconstruction, resulting in triangulated meshes. However, due to the dynamic nature of water, the feature points extracted from images taken at different moments and perspectives often lack consistent and mutual matches. Consequently, when using MVS for reconstruction, water bodies may exhibit a substantial number of gaps or holes. Additionally, surface reconstruction methods are not well-suited for handling multiple surfaces, particularly situations involving multi-layered object surfaces due to semi-transparency.

Supplementary Material.

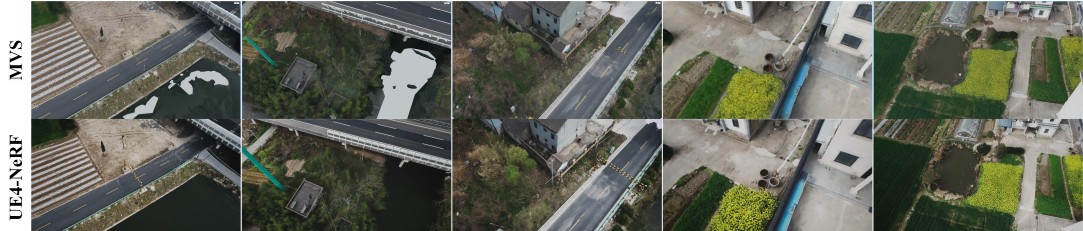

Figure 2: **Qualitative Comparison between MVS and proposed UE4-NeRF.** Pay attention to transparent objects and texture information.

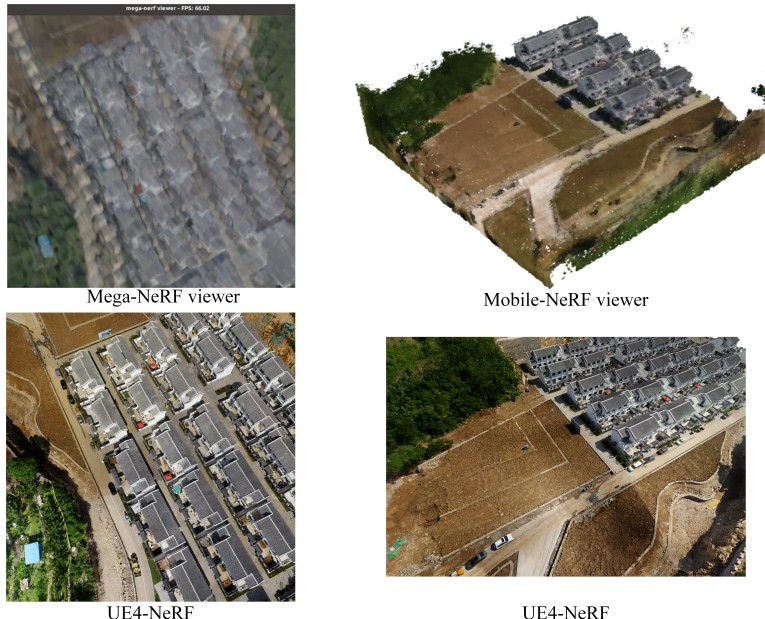

Mega-NeRF viewer                                    Mobile-NeRF viewer

UE4-NeRF                                             UE4-NeRF

Figure 3: **Qualitative comparison of different methods in the final renderer.** The final rendering effect in the renderer can truly compare the advantages and disadvantages of the methods.

Within the principles of MVS, there exists a parameter controlling the neighborhood range. Generally, the default neighborhood value prioritizes hole avoidance, which can result in suboptimal modeling effects for object detail structures and smaller objects. In MVS modeling, textured models are created by selecting an appropriate patch from all images to serve as the texture for a triangular face. Consequently, the color observed for this triangular face remains consistent from any angle during rendering. However, this approach works well only for objects that ideally adhere to the diffuse reflection model, particularly those that are opaque. In reality, some objects exhibit highlights and semi-transparency, and MVS-generated textures cannot accurately reflect these characteristics. As a result, rendering visual effects can be subpar. Our approach does not encounter the issues mentioned above.

## C  Qualitative comparison of different methods in the final renderer

In Figure!3, we can see that the results of UE4-NeRF in the renderer are much better than Mobile-NeRF and Mega-NeRF. Mobile-NeRF finally uses webgl for real-time rendering. We speculate that the poor rendering ability of webgl has a certain impact on the rendering quality. WebGL sorts objects according to the coordinates of the center of the object instead of sorting according to the surface, and there will be various strange interlacing. Real-time rendering results in Mega-nerf-viewer are weird.

# D  Storage overhead comparison during training

| (Construction Site) | Mega-NeRF | Mobile-NeRF | UE4-NeRF | Picture |
|---|---|---|---|---|
| Temporary Files | 158GB | 3.5GB(1/32 scene) | 19.2GB | 20.3G |

Table 1: **Storage overhead comparison during training.** Mega-NeRF generates a large number of temporary files.

# E  Gradient

In the paper, we have mentioned that the final loss function of the photometric part is computed as follows:

$$\mathcal{L}_{rgb} = \sum_{l=1}^{5} \beta \mathcal{L}_{rgb}^{part1}(\theta, V_p) + (1 - \beta) \mathcal{L}_{rgb}^{part2}(\theta, V_p) \tag{1}$$

The gradients obtained from the final backpropagation are used to update the weights ($\theta$) of the Encoder-Decoder network as well as the positions of the vertices $V_p$ in the polygonal meshes. As in Eq. 2, the gradients used to update the vertex positions are identical to the gradients obtained from the backpropagation of the overall photometric loss.

$$\nabla_{rgb}(V_p) = \nabla \mathcal{L}_{rgb} \tag{2}$$

*However, if all the gradients from different layers are initially used to update the network weights, it may cause the sparse polygonal meshes of higher layers to adopt colors before they have moved closer to the object surface. Furthermore, since the same network is used all the layers, it ultimately results in the dense vertices of lower layers to be unable to approach the object surface.* Taking these factors into consideration, we set different weights ($\gamma$) between the gradients of the recent layer and those of other layers.

$$\nabla_{rgb}(\theta) = (1 - \gamma) \nabla \mathcal{L}_{rgb}^{level-1} + \gamma \nabla \sum_{l=2}^{5} \mathcal{L}_{rgb}^{level-l} \tag{3}$$

$$\gamma = \begin{cases} 0, & \psi < 0, \\ \psi, & 0 < \psi < 0.5 \\ 0.5, & \psi > 0.5. \end{cases} \quad where \quad \psi = \lfloor \frac{epoch - 20000}{10000} \rfloor \times 0.17 \tag{4}$$

From Eq. 3 and Eq. 4, it can be observed that in the first 30,000 epochs, the gradients of all layers except the first layer are not used to update the network weights. Afterward, as the number of epochs increases, the weight parameter $\gamma$ gradually increases but does not exceed 0.5.

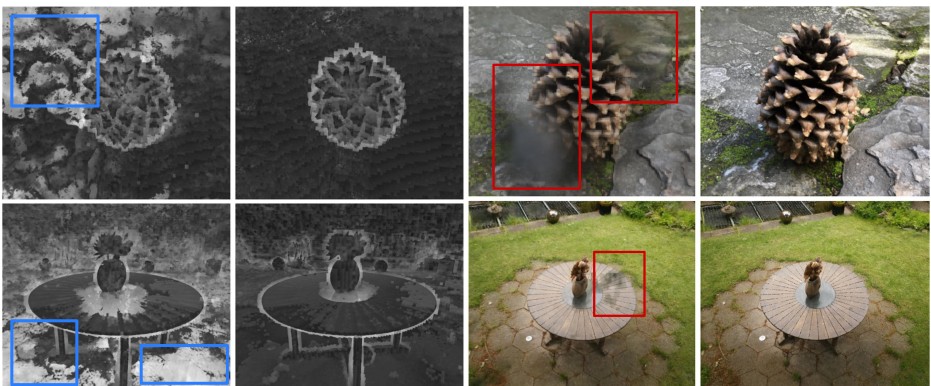

Figure 4: **Comparison between the usage and non-usage of pseudo-depth.** The first and third columns represent our results without incorporating pseudo-depth, while the second and fourth columns represent the results with pseudo-depth.

## F   Pseudo-depth

How to speed up the convergence and improve the quality of radiation field reconstruction is one of the most critical issue in NeRF. Instant-NGP shows that by using multi-resolution hash coding and optimizing the sampling structure, the training time can be reduced to several minutes, and the modeling scene can be rendered in real time using volume rendering. ***However, there are still artifacts, floating objects and time-consuming issues in the scenes reconstructed by instant-NGP.*** To tackle these, we propose to use the pseudo-depth as the supervision information of NeRF to speed up the convergence speed and improve reconstruction quality of the radiation field. The use of pseudo-depth supervision can increase the opacity ($\alpha$) value of the sampling points on the surface of the object, thereby greatly reducing the time required for rendering. ***As described in the main paper, pseudo-depth is obtained by calculating the distance between the sparse point cloud and the camera position.***

Feature points are generally points with translation, rotation and certain scale and brightness invariances. The feature points extracted from the image are very sparse. In addition, when estimating the camera pose, only a small part of these feature points can satisfy the multi-view constraints (reprojection error is less than the threshold), so the feature point cloud is very sparse. We find that the feature points occupy about 1/3000 of the pixels of the whole picture. For deep supervision, the more feature points the better, but it will greatly increase the preprocessing time. In order to solve the problem of unstable loss convergence and to ensure that the overhead calculation is not increased, and low contribution of deep supervision due to the fact that the number of feature points is too scarce, we sample pixels near feature points using the Gaussian distribution. Eq. (5) is the Gaussian distribution function. Since the surface of the object is continuous, it can be assumed that in most cases the distance between the point on the surface of the object near the feature point and the camera changes. Therefore, the Gaussian distribution function that follows to this trend is used as the confidence to describe a point that conforms to this assumption.

$$\omega_{ri} = \exp\left(-\frac{(x - x_i)^2}{2f}\right)\exp\left(-\frac{(y - y_i)^2}{2f}\right) \tag{5}$$

where $\omega_{ri}$ is the confidence of the pixel depth covered by a feature point. The threshold of $\omega_{ri}$ is 0.01. When $\omega_{ri} \leq 0.01, \omega_{ri} = 0$. $f$ is the scaling factor, which ensures that the pixels in the area covered by all feature points occupy about 6% of the entire image. It has been verified that the sampling points occupying 6% of the entire picture can achieve a good supervision effect and the overhead is very small. When the image pixels are $1600 \times 1600, f = 1$. $(x_i, y_i)$ is the coordinate of the feature point.

$$\omega_r = \begin{cases} \sum_{i=1}^{n}\omega_{ri}, & \sum_{i=1}^{n}\omega_{ri} \leq 1, \\ 1, & \sum_{i=1}^{n}\omega_{ri} > 1. \end{cases} \tag{6}$$

$$D(r) = \frac{\omega_{r1}D_{r1} + \omega_{r2}D_{r2} + \cdots + \omega_{rn}D_{rn}}{\omega_r} \tag{7}$$

where $D_{ri}$ is the pseudo-depth value of the feature point. For the cases where a pixel can be covered by multiple feature points, we use Eq. (7) to weight the depth of each feature point and use Eq. (6) to ensure that the sum of the weights of each depth does not exceed 1. $D(r)$ is the final pixel pseudo-depth value used for supervision.

We store pixel coordinates in **1D** array, and store depth map ($D(r)$) and weight map ($\omega_r$) in a HashMap. ***Compared with uncompressed storage (i.e., in a matrix method), storing in HashMap can reduce the memory utilization by more than*** $90\%$.

$$\mathcal{L}_D = \sum_{r \in \mathcal{G}} \omega_r \left\| \hat{D}(r) - D(r) \right\|_2^2 \tag{8}$$

Eq. (8) is the depth loss. Where $\mathcal{G}$ is the set of rays for which $\omega_r$ is non-zero in each batch. The predicted depth value is obtained by volume rendering:

$$\hat{D}(r) = \sum_{i=1}^{N} T_i \alpha_i t_i, \quad T_i = \prod_{j=0}^{i-1} (1 - \alpha_j) \tag{9}$$

where $t_i$ is the distance from the camera origin to the sampling point on the ray.

**Qualitative comparison.** Our pseudo-depth related experimental results are shown in Figure 4. The images are sourced from the publicly available object-based datasets[1]. In the comparison of depth maps, the pseudo-depth supervision significantly reduces the artifacts present in the depth maps which are indicated by the areas highlighted with blue bounding boxes. Additionally, it effectively mitigates the occurrence of floating objects in the scene, as indicated by the areas highlighted with red bounding boxes. The use of pseudo-depth greatly enhances the quality of rendering, and computing pseudo-depth does not incur significant additional computational costs.

# G   Illustration of some details

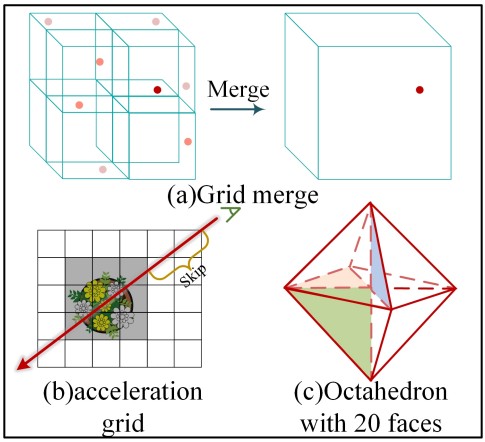

Figure 5: **Illustration of some details.** (a)Eight grids are merged into one to generate meshes for the rough layer. (b)Acceleration grid is used to skip grid and reduce calculations. (c)A regular octahedron has 20 faces in total, including the 8 exterior faces and 12 interior faces.

---

[1]nerf_real_360 dataset and 360_v2 dataset

## H   Transient Objects

We employ a segmentation network to generate masks for moving/dynamic objects and conduct comparative experiments on the rendered scenes before and after applying these masks. Figure 6 demonstrates the improvement in reconstruction quality achieved by using masks for moving objects. The first row in Figure 6 presents a comparison of depth maps, revealing the presence of holes in the depth map before using masks. These holes are caused by moving vehicles on the highway, resulting in inconsistent depth information on the road surface. After applying masks, the road surface depth becomes smooth and consistent. The second row in Figure 6 illustrates a comparison of the final rendering results. In scenes without masks, there are artifacts such as floating objects in certain perspectives.

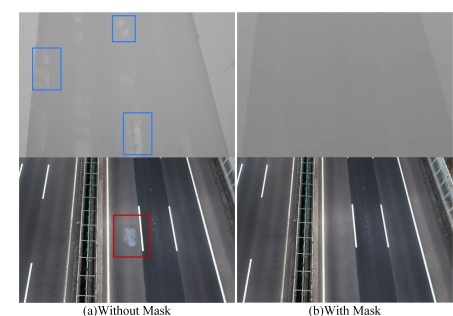

(a)Without Mask    (b)With Mask

Figure 6: **Comparison between the usage and non-usage of mask.** The use of masks can enhance the quality of rendering.

## I   Transparent and Semi-transparent Objects

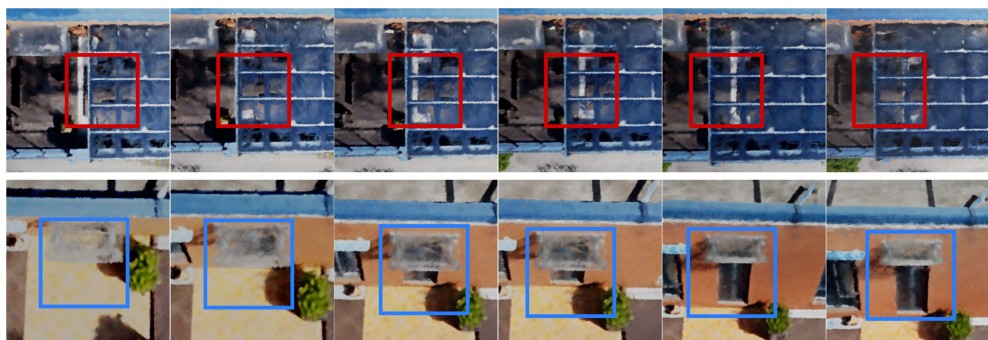

Figure 7: **Our results for transparent objects.** The transparent objects from different viewpoints are presented.

We employed the alpha-dithered technique to achieve rendering of transparent objects. As illustrated in Figure 7, we can perceive the objects located behind the transparent glasses which are annotated with bounding boxes. However, upon observing the glass in the right portion of the first column in Figure 7, it can be noticed that the transparency effect is not as prominent. This is due to the presence of dirt on the glass surface.

**Water.**   Water is one of the most common transparent objects in nature, and rendering water poses a challenging problem. In Figure 8, we showcase the rendering of water, which achieves high fidelity compared to the Ground Truth. Additionally, upon closer examination of the water surface, the rendering effectively captures the reflection on the water surface. To reduce the rendering cost of water, we only utilize the coarsest mesh when rendering the portion of water below the ground surface.

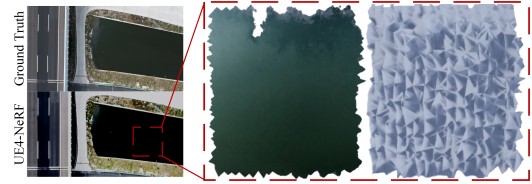

Figure 8: **Rendering of water.** Realistic reflection effects are achieved when rendering water.

## J   Mesh Conformity

Figure 9 shows meshes of the ground plane in the scene. It can be noted that most of the meshes are concentrated on the surface, while only a small portion of the meshes are present below the ground. In the final rendered scene, the thickness of the ground surface is not very significant; it is merely a thin layer. Figure 10 shows the internal structure of the building. The red line in the leftmost image represents the direction of subsequent motion. The meshes show the interior of the building as empty, with the meshes compressed onto the surface of objects. The loss function we designed and the pre-rendering process effectively control the meshes used for rasterization to be within the objects surface.This significantly reduces storage and computational costs.

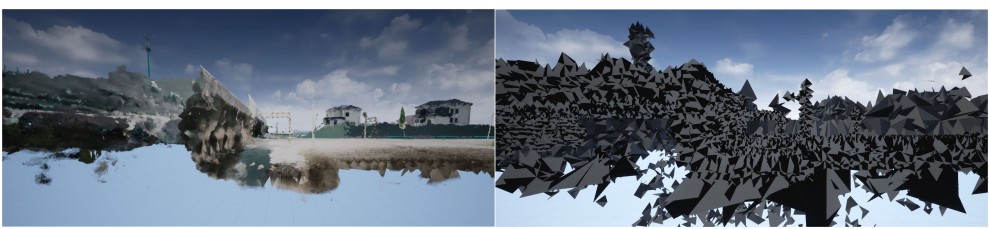

Figure 9: **Visualization of the ground plane of the scene terrain.** The left side displays the final rendered scene, while the right side shows the untextured mesh. It can be observed that below the ground surface, there are no chaotic scenes.

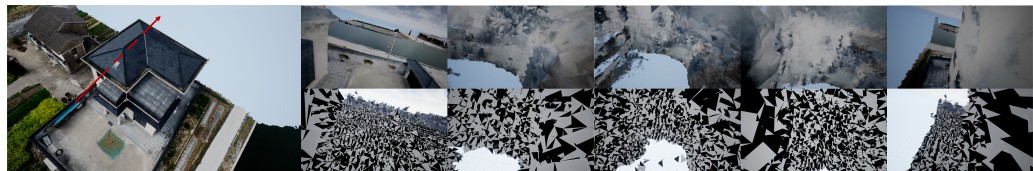

Figure 10: **Distribution of the meshes inside the buildings.** The interior of the rendered scene as well as the untextured meshes is empty. The meshes are compressed onto the surface of the objects.

# K  Mip-NeRF

By increasing the number of training iterations, we aim to assess whether the inadequate rendering quality of Mip-NeRF is attributed to insufficient training epochs. As illustrated in Figure 11 the horizontal axis represents the number of training epochs. We progressively increased the number of training epochs from the original 80,000 to 160,000, 240,000, 400,000, 560,000, and finally 800,000. However, when compared to the ground truth, Mip-NeRF still suffers from significant blurring artifacts. We also present the rendering results of UE4-NeRF after training for 80,000 epochs, showcased in the last row.

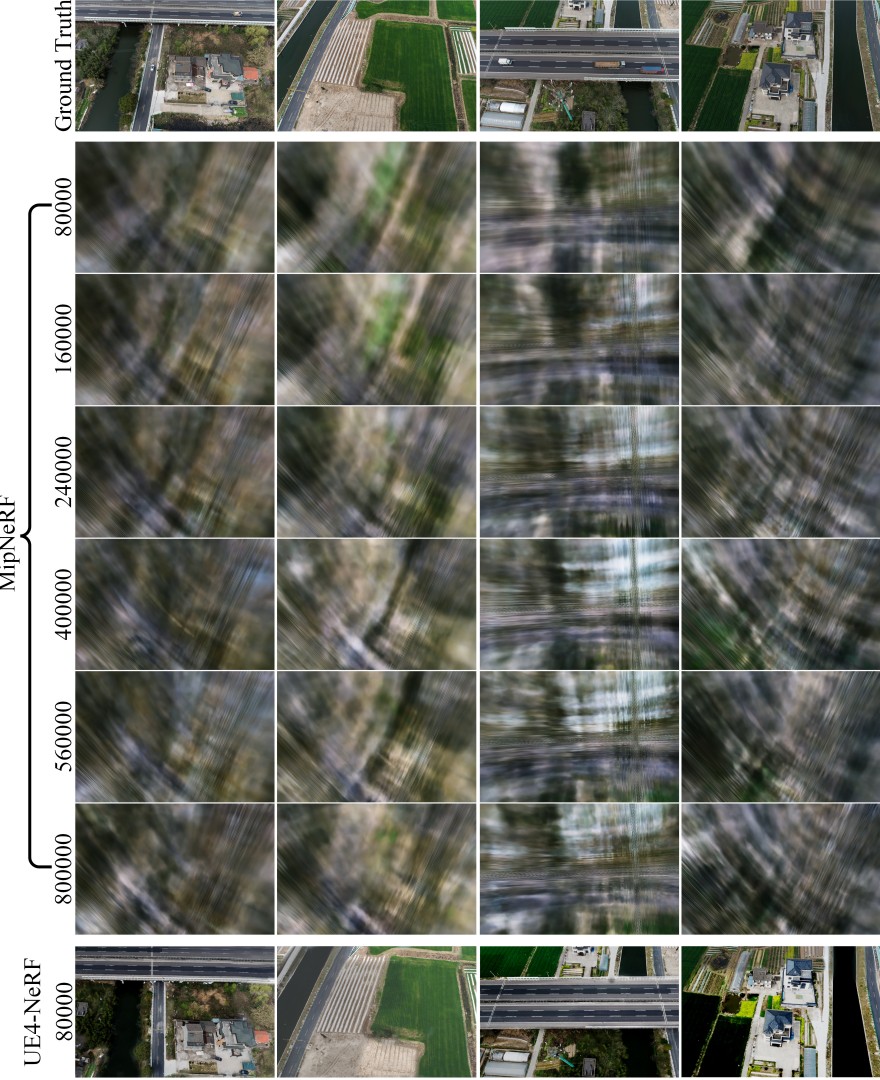

Figure 11: **The rendered outcomes vary with an increasing number of training epochs.** Despite increasing the training iterations of mipNeRF up to 800,000, the obtained results are still not satisfactory or as desired.

# L   Societal Impacts

Utilizing UE4-NeRF for real-time rendering of large-scale scenes may introduce potential social implications, including: 1) Digital privacy concerns. To enable real-time rendering of large-scale scenes, a substantial amount of photo data needs to be collected through unmanned aerial vehicles (UAVs) during the training process. This data may contain sensitive information, including facial attributes and personal identities. In cases of data misuse or inadequate protection, the risks of privacy breaches and potential exploitation increase, consequently amplifying the exposure of individuals' privacy; 2) Unrealism and illusion. NeRF's rendering capability can generate highly realistic 3D scenes, but this can also lead to difficulties in discerning reality from fiction, thereby influencing people's judgment and decision-making. 3) Threat to real-world environments and cultural heritage. Real-time rendering of large scenes enables the simulation and reconstruction of actual cities, buildings, and cultural heritage. However, this may result in reduced reliance on real-world environments and weaken the preservation and inheritance of cultural heritage.