# OpenReview forum: "UE4-NeRF:Neural Radiance Field for Real-Time Rendering of Large-Scale Scene"
_NeurIPS.cc/2023/Conference — NeurIPS 2023 poster_

### Official Review · Reviewer_3kud · 2023-07-06

**Soundness:** 3 good
**Presentation:** 3 good
**Contribution:** 3 good
**Rating:** 5
**Confidence:** 5

**Summary:**

The paper proposed a method for large-scale scene reconstruction and rendering.
It utilizes NeRF to learn a mesh representation for the scene by optimizing the
vertex positions and neural features/MLPs using standard volume rendering.
To handle large-scale scenes, the method divides the scene into multiple blocks
and trains a NeRF for each block. The initial mesh within each grid is got
by creating an octahedron at the center of each voxel grid. The paper also learns
a hierarchical representation with Level-of-Details to support efficient
rendering viewed from different distances. The paper shows that such a representation
can be integrated with Unreal Engine for interactive rendering and scene composition.
The paper shows results on outdoor scenes captured by drones and makes comparisons
against baselines such as instant-ngp, urban-NeRF, and NeRF-w. The experiment results
show that the proposed method achieves higher accuracy in rendering and faster rendering
speed.

**Strengths:**

1. The pipeline of optimizing a mesh representation with volume rendering for large-scale
scene reconstruction is technically sound, and the paper shows that the proposed pipeline
achieves better accuracy than the baseline method in terms of both accuracy and rendering speed.

2. The paper introduces multiple techniques to increase the rendering efficiency and quality
of the reconstructions, such as using an alpha threshold to make the mesh close to the real
surface, using pseudo-depth to remove floaters, and using pre-rendering to filter out
triangles that have small contributions.


**Weaknesses:**

See Questions.

**Questions:**


1. The overall pipeline of the paper is similar to that of MobileNeRF. The paper says in Line 100
that "in this work, we accelerate the rendering by ... into a mesh representation for efficient
mesh rasterization pipeline".  How is this different from MobileNeRF?
While I agree that the proposed method is faster than MobileNeRF in terms of training, I think
it's the main benefit of using hash grid instead of MLP. In terms of representations, I think
the proposed method is similar to MobileNeRF.
In terms of supporting large-scale inputs, it seems that the major change is separating the scene into blocks and doing blockwise
training. Can MobileNeRF be adopted for this task if we train it for each block? I think the paper
should make it more clear the difference of the proposed representation.

2. In Line 137, I think it should be "a regular octahedron with **8** faces". The paper says the
MobileNeRF has slow and unstable convergence issues. It's not clear to me why using octahedron
can help resolve this issue. What are the benefits of using it? It's needed to add a validation
study and comparisons to MobileNeRF. It will give the readers more insight if the paper
involves a comparison to MobileNeRF on standard benchmarks such as NeRF synthetic and LLFF.

3. The paper mentions having overlaps between blocks. How are the reconstructions handled at
overlapped regions? More details should be provided in this part.

4. In Equation 5, how is the threshold 0.3 chosen? How does the performance of the proposed method
change with different values of this threshold?

5. In Line 222, the paper says "filter out meshes that have an intersection opacity less 0.3". Is
this opacity only calculated at a single intersection point? I am wondering whether this will
cause problems in regions where there are sharp edges with abrupt opacity changes.

6. I think the title of the paper is kind of misleading. The reconstructed models are compatible
with standard rasterization engines using customized shaders, and it's not specific to UE4. Integrating
into UE4 is also something trivial given the underlying polygonal meshes. I would suggest the authors
modify the short name to more focus on large-scale reconstructions.


Overall, I think the proposed method is technically sound, and the presented results are convincing.
My major concern is that the backbone used in the paper is similar to MobileNeRF and the paper is combining
it with blockwise reconstructions (which is explored in previous large-scale NeRF works),
which prevents me from giving a higher rating.


**Limitations:**

The limitations look good to me.

---

> ### Author Rebuttal · Authors · 2023-08-10
>
> A1: Thanks for your comments. Actually, our proposed method is different work from Mobile-NeRF. **For a detailed comparison, please refer to Author Rebuttal.** In our test, using NGP in the same block requires 70,000 epochs to achieve results close to our method. Mobile-Nerf training is divided into three phases, the first two of which involve the training of the model, which requires 300,000 epochs for the first phase and 500,000 epochs for the second phase. Assuming we use a hash mesh instead of mobile-nerf's backbone, consider that neural network fitting opacity (high frequency information) requires more training than fitting volume density and requires additional mesh optimization. Then we can roughly estimate that the first stage needs at least 70,000 epochs to achieve the effect close to our method. Based on the highest speedup achieved in the first phase of NGP, the replaced mobile nerf backbone network needs at least 120,000 epochs to complete the second phase of training, and then in the third phase, Mobile nerf exports precomputed feature maps and grids. Our method only needs a total of 80,000 epochs to complete the training and export the UV coordinates of the vertices and mesh.
>   **In the Author Rebuttal, we provided a comprehensive comparison between UE4-NeRF and Mobile-Nerf.** Based on our previous experiments, training a block with Mobile-NeRF takes around two days on 4 * 3090Ti GPUs. Our focus is on modeling large scenes, and if we were to model a scene as described in the manuscript, it would require around two months. Additionally, Mobile-NeRF demands significant GPU memory during training, and the generated faces and features necessitate substantial storage. It's not feasible to load such a large number of models for real-time rendering.
>
> A2: A regular octahedron consists of a total of 20 faces, which includes the 12 interior faces. We attempted Mobile-NeRF's grid layout (initially without tilted surfaces), but found that it required more training epochs to achieve a fitting result on tilted roofs (An additional 20,000 epochs are needed compared to 80,000 epochs). Moreover, the rasterization artifacts were noticeable in certain shrub areas (it took an additional 50,000 epochs to significantly reduce the visible artifacts). On the other hand, our octahedral grid handles more complex face-to-face intersections. The average distance from any point in space to the nearest triangular face is smaller than in Mobile-NeRF's grid layout. This results in faster convergence in the vertex optimization part of our approach.
> The focus of our work is on images captured by drones in large-scale scenes with GPS information. One of our goals is to establish a measurable NeRF model for these scenarios, which requires GPS data for real-world scale conversion. We performed a coordinate system transformation prior to modeling.
> In the UE4 coordinate system, the positive direction of the X-axis points to the east, the positive direction of the Y-axis points to the south, and the positive direction of the Z-axis points up. The scale is 100:1. Compared with other NeRF models, it is also very meaningful and novel to be able to measure the actual scale in the scene modeled by NeRF.
> Additionally, the current version of our method relies on GPS data for various processing steps such as block division, optimal modeling height calculation, and ground estimation. However, existing public datasets lack GPS information within their images.  In principle, if we remove the reliance on GPS information, our approach is feasible for a variety of datasets. We are in the process of exploring the generalizability of the proposed approach tailored for aerial data, to be applicable across diverse scenarios.
>
> A3: When dividing the area, we accurately divide the scene into multiple areas, and there is no intersection between the target modeling areas of each area, but when modeling, in order to avoid the influence of the area mask error, the actual modeling the area is expanded outward by 1.333 times, but when exporting, we do not export the mesh of the expanded area.
>
> A4: The higher the threshold, the more the final model tends to capture opaque objects, while a lower threshold performs better in modeling translucent objects, but it results in more triangles during export, leading to increased rendering and storage costs. After comparing visualization effects and triangle counts across various threshold values, we found that a threshold of 0.3 strikes a good balance between effective modeling of translucent objects and reasonable triangle count.
>
> A5: Thanks for your reminder. A triangular face will only be clipped if the opacity of all intersection points on that face is below 0.3. In our rendering process, we haven't defined a minimum opacity threshold, and any valid opacity on effective triangular faces is preserved during asynchronous feature map computation. There's no threshold truncation. Hence, in practical rendering scenarios, this issue is scarcely encountered.
>
> A6: Currently, a significant portion of our related work is built upon the UE engine, encompassing features such as dynamic model loading, asynchronous feature map inference, and LOD layer calculations. Our rendering process is not a static loading process similar to mobile-nerf's static inference of feature maps and saving Mesh grids, and then the rendering engine loads for rendering, but real-time on-demand inference of feature maps and switching of LOD layers in rendering. The model is a dynamic process of predicting the model area that needs to be loaded and unloaded, and releasing data such as textures in the memory and cache when necessary. A large number of operations involve the implementation of specific APIs that depend on the UE4 engine. While theoretically adaptable to rendering engines like Unity, we have not been experimentally validated in other engines. Hence, in the interest of rigor, we refer to our approach as "UE4-Nerf."

---

> ### Author Response · Authors · 2023-08-21
> **Dear Reviewer 3kud, we are looking forward to your response.**
>
> Dear Reviewer 3kud, thank you for your concerns. In response to your question, we elaborated on the differences with Mobile-NeRF and supplemented our experiments. We apologize for the inconvenience of reaching out again, but as the deadline for the discussion phase is approaching, we find it necessary to send this reminder. ***We are anxiously awaiting your response and looking forward to further dialogue.***
>
> **Best regards and thank you once again.**

---

### Official Review · Reviewer_eVeu · 2023-07-07

**Soundness:** 3 good
**Presentation:** 2 fair
**Contribution:** 3 good
**Rating:** 5
**Confidence:** 5

**Summary:**

The authors introduce a multi-scale surface based representation for radiance field reconstruction and real-time rendering of large-scale scenes. The method involves subdividing the scene into partitioned scenes initialized with a regular octahedron mesh. Through joint optimization of the vertices' positions and multi-resolution hash grids, the proposed method incorporates two rendering loss functions and depth supervision. Experimental results demonstrate that the proposed method achieves improved rendering quality while maintaining real-time rendering capabilities (>30 FPS) for large-scale scenes at 4K resolution.

**Strengths:**

1. The manuscript successfully tackles the crucial challenge of fast training and rendering for large-scale scenes. Excellent work!

2. The multi-scale meshes representation seems fresh and interesting.

**Weaknesses:**

1. Presentation. The "transient object" section distracted me a lot, I don't this manuscript work on the dynamic scenario, correct me if I am wrong.

2. Evaluation. The proposed method is only evaluated on the UAV dataset, leading it hard to judge its general performance, would be great to have some results on standard dataset such as mipnerf 360 and Tanks and Temple.

3. Comparison. I could not find any comparison or discussion regarding Mobile-NeRF,  even though it is the closest method to the approach presented in this manuscript.

4. Fair comparison. Table 1 shows the proposed method provides clearly better performance than previous Mip-NeRF, NeRF-W NeRFacto and Instant-NGP, however, these methods don't use depth supervision by default, did you plug in the additional loss for these methods or use their official setting?

5. Anonymous. I don't think putting the project link associated with your personal GitHub account to the manuscript is a good idea. I'm not sure if it violates the anonymity rule, but I don't take that into account when scoring now and bring it up in hopes of hearing the authors' and another review's perspectives.

Minor:
The caption of Fig. 2: we partition the ...


**Questions:**

1. L137, why tilted surfaces can result in unstable convergence while regular octahedrons wouldn't?

2. The usage of "epochs" is confusing and not clear to me what is it? Do you want to refer it to "iterations"?

3. L196, how dense point clouds are used in the manuscript?

**Limitations:**

Yes, the authors provide a limitation section and make sense to me.

---

> ### Author Rebuttal · Authors · 2023-08-10
>
> A1. We can use masks to exclude dynamic objects (such as people or vehicles) and prevent them from affecting the rendering results.
>
> A2. Sincerely thank you for your suggestion.
> The focus of our work is on images captured by drones in large-scale scenes with GPS information. One of our goals is to establish a measurable NeRF model for these scenarios, which requires GPS data for real-world scale conversion.
> We performed a coordinate system transformation prior to modeling. We converted coordinates into a physically meaningful framework, establishing two transformations from NeRF coordinates to UE4's coordinates and GPS coordinates to UE4's coordinates.
> In the UE4 coordinate system, the positive direction of the X-axis points to the east, the positive direction of the Y-axis points to the south, and the positive direction of the Z-axis points up. The scale is 100:1. Compared with other NeRF models, it is also very meaningful and novel to be able to measure the actual scale in the scene modeled by NeRF.
> Additionally, the current version of our method relies on GPS data for various processing steps such as block division, point cloud generation from feature points, optimal modeling height calculation, and ground estimation. However, existing public datasets lack GPS information within their images.  In principle, if we remove the reliance on GPS information, our approach is feasible for a variety of datasets. We are in the process of exploring the generalizability of the proposed approach tailored for aerial data, to be applicable across diverse scenarios.
>
> A3. Based on our previous experiments, training a block with Mobile-NeRF takes around two days on 4 * 3090Ti GPUs. Our focus is on modeling large scenes, and if we were to model a scene as described in the manuscript, it would require around two months. Additionally, Mobile-NeRF demands significant GPU memory during training, and the generated faces and features necessitate substantial storage. It's not feasible to load such a large number of models for real-time rendering. Considering the rationale for comparison, we did not provide a complete comparison in the manuscript. **In our "Author Rebuttal", we discussed the comparison between our approach and Mobile-NeRF. Moreover, in the submitted PDF, we provided both qualitative and quantitative comparisons with Mobile-NeRF.**
>
> A4. In order to further speed up the convergence and reduce the abnormal phantom-like floating objects in the air, we propose pseudo-depth to optimize the training, which itself is an innovative method proposed for our scenario. Compared with others, their official code is used, but the pseudo-depth method we proposed is suitable for traditional NeRF methods, and we can try to apply pseudo-depth to the mentioned NeRF framework later. Your invaluable review and insights are greatly appreciated.
>
> A5. Thank you. While it is a private GitHub account, we removed all personal information before manuscript submission. This maintains anonymity and adheres to the principle of anonymity.
>
> A. 5.5 Thanks for your suggestion, we will modify it in the revised version.
>
> A6. The grid layout we use differs from that of Mobile-NeRF. A regular octahedron consists of a total of 20 faces, which includes the 8 tilted exterior faces and the 12 interior faces. We attempted Mobile-NeRF's grid layout (initially without tilted surfaces), but found that it required more training epochs to achieve a fitting result on tilted roofs (reaching a visual quality similar to our current approach required an additional 20,000 epochs of training, whereas our method only needed 80,000 epochs). Moreover, the rasterization artifacts were noticeable in certain shrub areas (although they improved with further training, it took an additional 50,000 epochs to significantly reduce the visible artifacts). On the other hand, our octahedral grid (including 12 interior triangular faces) handles more complex face-to-face intersections. The average distance from any point in space to the nearest triangular face is smaller than in Mobile-NeRF's grid layout. This results in faster convergence in the vertex optimization part of our approach.
>
> A7. I'm sorry for confusing you with my unclear expression. Yes, epochs refers to iterations.
>
> A8. The point cloud points here are derived from feature points with minimal matching loss during camera pose estimation. We employ them as sparse depth supervision targets. Given that this point cloud is extremely sparse, with a ratio of about 1/10000 compared to pixels, we use a certain neighborhood around the projected coordinates of the point cloud on the image as supervision targets and confidence diminishes with increasing projection distance, resulting in lower supervision strength. Specific implementation details can be found in the attached document.

---

> > ### Comment · Reviewer_eVeu · 2023-08-16
> > **additional results**
> >
> > Thanks for the rebuttal, it resolves my concerns.
> >
> > The previous main concerns from the reviewers are the incomplete comparisons and unclear exposition, given the authors provide some new results, would love to hear back from Reviewer x4aP and JaPB.

---

> > > ### Author Response · Authors · 2023-08-17
> > > **Thanks for your affirmation of our response.**
> > >
> > > Thanks for your affirmation of our response. We responded to your concerns in detail and added additional experimental results. Diagrams are in the submitted PDF.
> > > If you have any further concerns, please do not hesitate to tell us. Thanks again!

---

> ### Author Response · Authors · 2023-08-22
> **Dear reviewer eVeu, the Reviewer x4aP and JaPB you mentioned responded to our Rebuttal and improved the rating!**
>
> Dear reviewer eVeu,
>
> I am reaching out again because  reviewers you mentioned responded to our rebuttal and raised the rating. We  included additional comparative experiments with Mega-NeRF **on public datasets** as you suggested. Kindly review our discussion with reviewer x4aP. UE4-NeRF has demonstrated remarkable outcomes.
>
> Best regards and thank you once angin!

---

### Official Review · Reviewer_PLf2 · 2023-07-09

**Soundness:** 3 good
**Presentation:** 1 poor
**Contribution:** 2 fair
**Rating:** 5
**Confidence:** 4

**Summary:**

This work presents a system to represent a large scene using NeRF given drone-captured photos. It partitions a large scene into overlapping smaller tiles, and represent each tile with a sub-NeRF. To enable real-time rendering, it represents each sub-NeRF using meshes and neural textures (represented as a encoder-decoder network) during training; after training, meshes are further simplified, and the encoder part of neural textures are converted to texture image on request, while the decoder part is implemented as a GLSL shader. The whole real-time rendering system for the proposed NeRF representation is built on top of Unreal Engine 4, and allows for scene editing.

**Strengths:**

The proposed method not just trains fast, but also renders fast. It's cool to see the whole NeRF real-time rendering system integrated into the existing renderer, Unreal engine 4.

**Weaknesses:**

1. Writing could use some improvements; many key technical details seem missing, making the paper a bit hard to understand.
* 1.1 The frame rate of real time rendering is reported on what device?
* 1.2 Line 129-132 says that "we train a coarse model for sub-region segmentation..."; what model was trained here? Is it counted in the total training time of the proposed method?
* 1.3 Line 143 says "it generates 8D feature vector, which incorporates opacity information"; this seems very confusing to me, as in Eq. 1, the opacity is predicted alongside the feature vector.
* 1.4 Line 167-169 seems to suggest that max should be used in Eq. 5, as opposed to min?
* 1.5 Octahedron with 20 faces. Might be better to explain why there're 20 faces instead of regular 8 faces.
* 1.6 Fig 5 village scene, the lower-right inset of ours does not seem to align with those of baselines.
* 1.7 How is the comparison with baselines with MipNeRF and InstantNGP performed? Were the baselines trained on each individual blocks or they were trained on the whole scene?
* 1.8 Line 248-254: alpha values seem to be dropped during real-time rendering. If so, please make it explicit, and include examples how such dropping affects the final rendering quality.
* 1.9 Fig. 1c seems confusing without labels "before editing" and "after editing". In addition, how was the inpainting performed when the building/car were removed?
* 1.10 I find it hard to understand what's going on in line 181-185. The authors might want to add some illustrations.
* 1.11 Line 52-54 seems to suggest that the speed up of training, compared with MobileNeRF, is due to implement some compute-heavy components using customized cuda functions. What are these components?

2. Ablations and comparisons can be improved; it's hard to judge if the proposed components are useful or not, and how it compares against baselines.
* 2.1 The BungeeNeRF work seems to a better baseline than MipNeRF; but for some reason, comparison with that work is missing.
* 2.2 Why not compare with MobileNeRF on each tile? Line 135-137 claims that the proposed mesh initialization addresses the unstable convergence issue of MobileNeRF; is there any intuitive explanation or empirical evidence to support this claim?
* 2.3 Ablations are needed to justify the effectiveness of the loss function in Eq. 6.



**Questions:**

1. It seems that the proposed method is a bit complicated during training phase. Will the training code be released for reproducibility upon acceptance?

2. Would the proposed method also work for blender, besides the unreal engine 4?

**Limitations:**

Limitations are adequately addressed.

---

> ### Author Rebuttal · Authors · 2023-08-10
>
> A1.1 In UE4, you can input the command "stat fps" to show frame rate.
>
> A1.2 Training a NGP model on low-resolution images of the entire scene aids us in better segmenting the scene. This step takes only a matter of minutes.
>
> A1.3 Opacity is solely dependent on position and not influenced by direction. As a result, we can directly obtain opacity through the encoder. Utilizing the encoder, we derive an 8-dimensional feature vector, where one dimension's feature can be transformed into opacity via a nonlinear transformation. This is why we assert that the 8-dimensional feature vector encompasses opacity information.
>
> A1.4 Eq 5 is correct. The lines 170-173 should be revised to read: "In the initial 10,000 epochs, the f is maintained at 0., and afterward, the f increases as the number of epochs increases, but does not exceed 0.3."
>
> A1.5 A regular octahedron has 20 faces in total, including the 8 exterior faces and 12 interior faces. **We have included illustrative diagrams of three representative interior faces in the submitted PDF(Figure 3(c)) to aid your understanding.**
>
> A1.6 To facilitate accurate measurements with our established model, we performed a coordinate system transformation to make it have physical meaning. There exists a slight discrepancy between coordinates in UE4 and those estimated by COLMAP, but this error is minor and has negligible impact on the overall visual comparison. Currently, we have rectified this discrepancy and assure that in subsequent material revisions, adjustments will be made for consistency, enhancing the visual prominence.
>
> A1.7 They were trained on the whole scene.
>
> A1.8 We do not discard opacity values. The Alpha information is preserved within the 8 channels, and it is calculated non-linearly from one of those channels.
>
> A1.9 The first line is before editing, and the second line is after editing. We directly remove the triangular mesh from the area and copy a portion of triangular mesh from surrounding similar regions.
>
> A1.10 **We illustrate the generation of different levels of detail (LOD) in the submitted PDF (Figure 3(a)).** We choose the point in the grid with the largest opacity as the point of the synthesized grid to generate the octahedron, and the point on the edge is not used as the center point of the octahedron.
>
> A1.11 Component: UV sampling based on vertices, generating feature maps from UV-sampled points, followed by BC4 compression. The description of "therefore" in line 53 of the manuscript is not entirely accurate. The impact is not solely due to the implementation in CUDA; this is merely one contributing factor, albeit not the determining one. Our approach, even without utilizing CUDA implementation, achieves training times of less than 2 hours for a region on a single 3090Ti. With CUDA , this time is further reduced to less than 1 hour.
>
> A2.1 Thanks for your suggestion, and we also tried. But it's highly impractical to obtain a wide range of multiscale images, from low to high, for training BungeeNeRF. The data used in BungeeNeRF is derived from various scales of rendered images of models in Google Earth, rather than being captured from real-world imagery. **In the "Author Rebuttal", we extensively compared our approach with Mega-NeRF and Mobile-NeRF.**
>
> A2.2 Based on our previous experiments, training a block with Mobile-NeRF takes around two days on 4x3090Ti GPUs. Our focus is on modeling large scenes, and if we were to model a scene as described in the manuscript, it would require around two months. Additionally, Mobile-NeRF demands significant GPU memory during training, and the generated faces and features necessitate substantial storage. It's not feasible to load such a large number of models for real-time rendering. Considering the rationale for comparison, we did not provide a complete comparison in the manuscript. **In our "Author Rebuttal" and submitted PDF, we discussed the comparison between our approach and Mobile-NeRF.**
> 	The grid layout we use differs from that of Mobile-NeRF. We attempted Mobile-NeRF's grid layout (initially without tilted surfaces), but found that it required more training epochs to achieve a fitting result on tilted roofs (reaching a visual quality similar to our current approach required an additional 20,000 epochs of training, whereas our method only needed 80,000 epochs). Moreover, the rasterization artifacts were noticeable in certain shrub areas (although they improved with further training, it took an additional 50,000 epochs to significantly reduce the visible artifacts). On the other hand, our octahedral grid (including 12 interior triangular faces) handles more complex face-to-face intersections. The average distance from any point in space to the nearest triangular face is smaller than in Mobile-NeRF's grid layout. This results in faster convergence in the vertex optimization part of our approach.
>
> A2.3 **In the submitted PDF (Figure 5), we provide rendered results without the second part of the loss.** When not utilizing the second part of the loss, the model tends to utilize a multitude of low-opacity faces to fit in regions of low color variation and semitransparency. This leads to accidental deletion of faces that should be part of the model's surface when exporting triangles. In scenes with glass and shadows, more triangles may be unexpectedly deleted. This effect is especially prominent on glass surfaces in the picture, where numerous triangles may be unintentionally removed.
>
>
> A3. If our manuscript is fortunate enough to be accepted, we plan to make the training code publicly available!
>
> A4. It seems you also have a keen interest in computer graphics! Currently, much of our related work is based on the Unreal Engine. While our method is theoretically applicable to other rendering engines like Unity or Blender, we haven't conducted experimental validation in those environments. Once again, thank you for the questions and suggestions you've raised!

---

> ### Author Response · Authors · 2023-08-21
> **Dear Reviewer PLf2, we are looking forward to your response.**
>
> Dear Reviewer PLf2, thank you for your concerns. We apologize for the inconvenience of reaching out again, but as the deadline for the discussion phase is approaching, we find it necessary to send this reminder. ***We are anxiously awaiting your response and looking forward to further dialogue.***
>
> **Best regards and thank you once again.**

---

### Official Review · Reviewer_JaPB · 2023-07-10

**Soundness:** 3 good
**Presentation:** 2 fair
**Contribution:** 3 good
**Rating:** 5
**Confidence:** 3

**Summary:**

This paper presents a method that combines NeRF and the Unreal Engine for real-time rendering of large-scale scenes. The method first partitions large scenes into sub-blocks, and represent NeRF via polygonal meshes initialized from regular octahedron. The opacity and feature vector are represented via hash-encoding. The mesh, opacity, and features are optimized during training. Inspired by LOD, the method trains meshes with different levels of detail to improve rendering efficiency at different scales. The optimized mesh can be integrated into the Unreal Engine 4 to achieve real-time rendering of large-scale scenes. Experiments demonstrate that the proposed method achieves better rendering quality than existing approaches and supports real-time rendering.

**Strengths:**

- How to achieve real-time rendering of large-scale scenes under the NeRF setting is an important problem. This paper presents, to the best of my knowledge, the first solution to this problem. The main idea is to represent the NeRF via polygonal meshes so that they can be combined with the rasterization pipeline in Unreal Engine after optimization. Several other designs are introduced such as block partition and LOD representations. The overall design is reasonable.

- The experimental results are impressive. UE4-NeRF shows clearly better rendering quality than other baselines such as Mip-NeRF and Instant-NGP. It is also the only method that supports real-time rendering of 2k large-scale scenes.

**Weaknesses:**

- Some baselines are lacking.
  - The closest method to the proposed UE4-NeRF is Mobile-NeRF, as it also represents NeRF via polygonal mesh. However, the comparison to Mobile-NeRF is missing. According to line 53, UE4-NeRF is faster than Mobile-NeRF because it implements computationally intensive portions with CUDA. However, this advantage comes from different implementations rather than the method itself. Authors should compare the two methods in a fair setting to demonstrate the benefits of the techniques proposed in this paper.
  - I suggest authors to also compare with Mega-NeRF (Mega-nerf: Scalable construction of largescale nerfs for virtual fly-throughs), which also studies building NeRF for large-scale scenes.

- The writing lacks clarity. For example:
  - For the initialized mesh, line 10 mentions "octahedra" (i.e., 8 faces), line 55 mentions "tetrahedra" (i.e., 4 faces), while line 137 mentions "octahedron with 20 faces". These terms are inconsistent and make it very confusing which one is correct.
  - At Eq.(1), it is not explained what is $P_i$. Is it the 3D coordinate of the point of intersection? Please clarify.
  - At line 142 "encoded by multi-resolution hash functions", authors should mention that this follows Instant-NGP and cite it.
  - At line 208, what is "acceleration grid"?

- In Fig.5, not all baselines are included. Authors should provide the qualitative results of other baselines in the supplementary material. The ground truth is also missing.
Besides, I noticed that the camera poses for different methods are not exactly the same. The camera poses should keep consistent in qualitative comparison.

- Grammar:
  - Line 217: "we incorporate parallel lights from various angles are added above." There are two verbs in this sentence.

**Questions:**

- Authors may respond to my concerns in the weaknesses.

- At line 290, it is mentioned that "the actual rendering quality of UE4-NeRF is expected to surpass the metric’s performance, as achieving consistent exposure matching with the original image is challenging within the Unreal Engine environment." Then I suggest authors also report the performance using original volume rendering to remove the exposure gap.

**Limitations:**

Yes.

---

> ### Author Rebuttal · Authors · 2023-08-10
>
> A1: Thank you for your encouragement and advice. The description of "therefore" in line 53 of the manuscript is not entirely accurate. The impact is not solely due to the implementation in CUDA; this is merely one contributing factor, albeit not the determining one. Our approach, even without utilizing CUDA implementation, achieves training times of less than 2 hours for a region on a single 3090Ti. With the integration of CUDA implementation, this time is further reduced to less than 1 hour.
> Our training strategy diverges from Mobile-Nerf, and we optimize the method for generating polygonal meshes. **In  our submitted PDF and  Author Rebuttal, we provided both qualitative and quantitative comparisons with Mobile-NeRF.** There are several main reasons for the slow training speed and poor rendering quality of Mobile-NeRF:
> 1. The model backbone's inference is slow, resulting in slow training for individual batches. Additionally, each batch demands a substantial amount of GPU memory. With a constrained GPU memory size, further expanding the batch size is not feasible. Consequently, training necessitates a greater number of epochs to complete;
> 2. Slow convergence of the grid-like structure. We attempted Mobile-NeRF's grid layout (initially without tilted surfaces), but found that it required more training epochs to achieve a fitting result on tilted roofs (reaching a visual quality similar to our current approach required an additional 20,000 epochs of training, whereas our method only needed 80,000 epochs). Moreover, the rasterization artifacts were noticeable in certain shrub areas (although they improved with further training, it took an additional 50,000 epochs to significantly reduce the visible artifacts). On the other hand, our octahedral grid (including 12 interior triangular faces) handles more complex face-to-face intersections. The average distance from any point in space to the nearest triangular face is smaller than in Mobile-NeRF's grid layout. This results in faster convergence in the vertex optimization part of our approach.
> 3. Time-consuming separation of sampling points for triangular faces into strictly transparent and opaque regions. This process contributes significantly to training time.
> 4. The absence of depth information supervision also leads to slow convergence. In contrast, our approach benefits from sparse depth supervision, reducing the number of epochs required for training.
>
> A2: **In our submitted PDF and "Author Rebuttal", we have conducted a comprehensive comparison with Mega-NeRF.**
>
> A3: The description on line 55 is inaccurate and should be "octahedron" instead of " tetrahedra." A regular octahedron has 20 faces in total, **including the 8 exterior faces and 12 interior faces( In PDF Figure 3(c)).**
>
> A4:Yes!
>
> A5:We will further explain the hash function and cite instant-NGP in the revised version.
>
> A6:In the training architecture of "Instant-NGP", there exists a 128x128x128 binary dense grid structure designed to efficiently skip empty regions during rendering. We have leveraged this dense grid while calculating the visible grid, allowing for rapid culling of empty regions within the triangular mesh. **In the submitted PDF(Figure 3(b)), we illustrate this method.**
>
> A7:Thank you for your attention to this detail. In the revised version, we will provide qualitative results for other baselines in the supplementary material.
> To facilitate accurate measurements with our established model, we converted coordinates into a physically meaningful framework, establishing a transformation from GPS coordinates to UE4's coordinates. In the UE4 coordinate system, the positive direction of the X-axis points to the east, the positive direction of the Y-axis points to the south, and the positive direction of the Z-axis points up. The scale is 100:1. Compared with other NeRF models, it is also very meaningful to be able to measure the actual scale in the scene modeled by NeRF. However, There exists a slight discrepancy between coordinates in UE4 and those estimated by COLMAP, but this error is minor and has negligible impact on the overall visual comparison. Currently, we have rectified this discrepancy and assure that in subsequent material revisions, adjustments will be made for consistency, enhancing the visual prominence.
> To aptly showcase more intricate details from diverse perspectives, we have opted to compare images captured from free viewpoints rather than images from the training viewpoint. Consequently, the ground truth is missing.
>
> A8:Thanks for your suggestion, we will modify it to "we incorporate parallel lights from various angles above".
>
> A9:We responded to all weaknesses. Sincere thanks for all your suggestions!
>
> A10:Due to the significant presence of intersecting faces in our octahedral mesh, while it aids model convergence during training, it introduces complications in volume rendering. In our method, during step-wise rendering, a substantial number of triangle intersections and depth sorting are required before rendering can proceed according to the transparency blending formula. This results in slow rendering speeds when adopting a volume rendering approach. Additionally, the challenge of blending multiple models from various regions further adds complexity. Addressing this issue demands efforts akin to developing a dedicated rendering engine, which contradicts our initial intention of integrating rendering seamlessly into the UE rendering engine. One positive development is that we've successfully resolved the camera exposure issue in UE4 by replacing the tone mapper with a custom tone mapper during the post-processing phase of rendering. In the revised version, we will update our rendering results. **In the submitted PDF, the presented results for UE4-NeRF depict the outcomes achieved after resolving the exposure gap, revealing an observable improvement in rendering quality.**

---

> > ### Comment · Reviewer_JaPB · 2023-08-17
> > **Replying to rebuttal**
> >
> > Thank you for your response. The rebuttal has addressed most of my concerns. So I raised my rating accordingly.

---

> > > ### Author Response · Authors · 2023-08-17
> > > **Thank you for your affirmation of our response!**
> > >
> > > Thank you for your affirmation of our response. Improving the rating gave us great encouragement! If you have any further concerns, please do not hesitate to tell us. Thanks again!

---

### Official Review · Reviewer_x4aP · 2023-07-14

**Soundness:** 2 fair
**Presentation:** 2 fair
**Contribution:** 2 fair
**Rating:** 5
**Confidence:** 4

**Summary:**

This paper introduces UE4-NeRF, a system that combines Neural Radiance Field (NeRF) with the Unreal Engine 4 (UE4) for real-time rendering and editing of large-scale 3D scenes. To achieve this, the system partitions scenes into sub-NeRFs and represents them using optimized polygonal meshes based on regular octahedra and tetrahedrons. Leveraging a Level of Detail (LOD) approach and the powerful development capabilities of UE4, the system enables high-performance rendering at different observation levels and seamless scene editing. Experimental results show that UE4-NeRF achieves rendering quality comparable to state-of-the-art methods. and also  accelerates the training speed.

**Strengths:**

- This paper describes a method that addresses the scalability limitations of NeRF by dividing the scene into smaller chunks and enabling parallel processing. This allows for the construction of large-scale NeRF models with reduced memory requirements and increased computational efficiency.

**Weaknesses:**

- Overall results: The results present an aerial view of an urban scene with good resolution. However, it is not clear how the authors leverage NeRF's capabilities in this scenario. Specifically, a comparison with traditional Multi-View Stereo (MVS) should demonstrate view-dependent effects, subtle details, transparent objects, and how it performs compared to multi-view methods. Additionally, it would be beneficial to include a first-person wandering case similar to Urban NeRF[32].

- Contribution: From my understanding, this method combines previous work on Urban NeRF[32] scene partitioning with Instant-NGP. While it has its merits, compared to Instant-NGP, the qualitative improvement in the results is marginal overall.

- Experiments: The authors have compared their method with their own dataset, which is commendable. However, it is encouraged to include comparisons with established datasets such as the one used in Mega-NeRF[40]. Additionally, qualitative and quantitative comparisons with Mega-NeRF[40] would provide further insights.

**Questions:**

see above

**Limitations:**

Based on the concerns raised, the marginal qualitative improvement compared to Instant-NGP, lack of comprehensive comparisons with established datasets, and unclear demonstration of NeRF's capabilities in the urban scene, it is recommended to borderline reject the paper unless relatively significant revisions are made.

---

> ### Author Rebuttal · Authors · 2023-08-10
>
> - **In the submitted PDF (Figure 4), we have provided additional qualitative comparisons with MVS.** Our experiments offer comparisons in transparent objects as well as subtle details. MVS utilizes sparse reconstruction to extract feature points, which are then expanded based on morphological and color differences to generate a dense point cloud. This dense point cloud is further used for surface reconstruction, resulting in triangulated meshes. However, due to the dynamic nature of water, the feature points extracted from images taken at different moments and perspectives often lack consistent and mutual matches. Consequently, when using MVS for reconstruction, water bodies may exhibit a substantial number of gaps or holes. Additionally, surface reconstruction methods are not well-suited for handling multiple surfaces, particularly situations involving multi-layered object surfaces due to semi-transparency.
>
>   Within the principles of MVS, there exists a parameter controlling the neighborhood range. Generally, the default neighborhood value prioritizes hole avoidance, which can result in suboptimal modeling effects for object detail structures and smaller objects. In MVS modeling, textured models are created by selecting an appropriate patch from all images to serve as the texture for a triangular face. Consequently, the color observed for this triangular face remains consistent from any angle during rendering. However, this approach works well only for objects that ideally adhere to the diffuse reflection model, particularly those that are opaque. In reality, some objects exhibit highlights and semi-transparency, and MVS-generated textures cannot accurately reflect these characteristics. As a result, rendering visual effects can be subpar. Our approach does not encounter the issues mentioned above.
>
>    Urban-NeRF is trained on a dataset of horizontally captured street views, making it most suitable for simulating first-person wandering along trekker motion routes. In contrast, our training images are taken from a top-down drone perspective, representing an aerial viewpoint. This aerial perspective aligns with the first-person wandering view in our application. Our project's demonstration video showcases first-person aerial wandering. Additionally, we have included our rendering plugin code and model, allowing users to control camera poses and perform real-time rendering independently.
> - Thanks for your comments. Actually, our proposed method is quite different work as combination of Urban NeRF and NGP. The main differences are as followings:
> 1. Distinct methods of scene representation. We trained meshes of multiple scales to represent a scene. And effectively combined with the LOD system in UE4.
> 2. The rendering principle is different. In UE4, the high-quality feature texture map on the mesh surface is asynchronously inferred and the MLP inference in the real-time rendering pipeline is combined to achieve high-resolution real-time rendering. However, NGP cannot be integrated into rendering engines such as UE4 for use, nor can it be conveniently added to the modeling scene for mixed rendering.
>
>    **Please refer to "Supplementary Explanation of NGP and Block Strategy." in "Author Rebuttal" for further discussion**. Through our experiments, simple combination of NGP and block strategy cannot achieve high quality and high frame rate rendering. We just combine the hash grides to minimize the training time for a single epoch. And in large scenes, if real-time rendering is considered, partitioning strategy is basically the only option given limited hardware storage and computing resources. The focus of our paper is not the combination of the choice of block strategy and a certain nerf training method, but the rapid construction of each block area on a single RTX3090-level graphics card in less than 1 hour, and the ability to achieve high resolution Rendering in real time.
>
>     Our method uses multi-scale meshes to represent the scene and greatly optimizes the rendering process, achieving high rendering accuracy, fast speed, and high frame rate. We managed to attain 50 fps at 2K resolution using a single 3090Ti. Moreover, as we further increased the screen resolution to 4K, the main impact was just on the frame rate, with a relatively minor increase in VRAM usage. In summary, our approach extends beyond a mere amalgamation of NGP and block-wise strategies. We exhibit discernible advantages in rendering scene scale, quality, speed, memory utilization, and other relevant aspects.
>
>
>
>
> - Thank you for your encouragement and advice. **In submitted PDF and "Author Rebuttal", we conducted a comprehensive comparison with Mega-NeRF**. The focus of our work is on images captured by drones in large-scale scenes with GPS information. One of our goals is to establish a measurable NeRF model for these scenarios, which requires GPS data for real-world scale conversion. We performed a coordinate system transformation prior to modeling. In the UE4 coordinate system, the positive direction of the X-axis points to the east, the positive direction of the Y-axis points to the south, and the positive direction of the Z-axis points up. The scale is 100:1. Compared with other NeRF models, it is also very meaningful and novel to be able to measure the actual scale in the scene modeled by NeRF. Additionally, the current version of our method relies on GPS data for various processing steps such as block division, point cloud generation from feature points, optimal modeling height calculation, and ground estimation. However, existing public datasets lack GPS information within their images.  In principle, if we remove the reliance on GPS information, our approach is feasible for a variety of datasets. We are in the process of exploring the generalizability of the proposed approach tailored for aerial data, to be applicable across diverse scenarios.

---

> > ### Comment · Reviewer_x4aP · 2023-08-20
> >
> > I have reviewed the comments from other reviewers as well as the author's response. I appreciate the author's efforts in conducting additional experiments with MVS and Mega-NeRF. However, for the Mega-NeRF experiment, the author opted for their own dataset instead of the established public dataset provided by Mega-NeRF. This choice makes it challenging for readers to evaluate the results with confidence. While I recognize the author's aim to highlight the uniqueness of their data's GPS information, it remains meaningful to perform experiments on a public dataset. I'll await the author's response to this point before making my final decision.

---

> > > ### Author Response · Authors · 2023-08-21
> > > **Dear reviewer, we supplemented our experiments on public datasets.**
> > >
> > > Thank you for the suggestions you provided. We incorporated additional comparative experiments using the dataset introduced in Mega-NeRF, and included links to the comparative result images in each subheading.
> > >
> > > ### [Comparison with Mega-NeRF](https://github.com/JamChaos/UE4-NeRF/blob/master/pictures/ours_mega_nerf_compare.jpg):
> > > We conducted qualitative comparisons with Mega-NeRF using the dataset it provided. We employed the pre-trained model for the "building" scene provided by Mega-NeRF's authors, which was divided into 8 sub-regions. Given that the scene provided by the authors is close to a square shape, following the blocking strategy of UE4-NeRF, we partitioned the "building" scene into 3x3 , which aligns closely with the blocking scale of Mega-NeRF. We performed visual comparisons within  final real-time renderer and  Mega-NeRF employs dynamic rendering to enhance rendering quality. Additionally, Mega-NeRF's dataset lacked controlled camera exposure and suffered from lengthy capture times, resulting in ambiguous observations of ground shadows from different viewing angles. Consequently, our approach exhibits some gaps in certain ground shadow areas and occasional anomalies such as artifacts that appear to float. **Despite these challenges, the final comparison is still striking. Our rendering quality significantly outperforms that of Mega-NeRF. Moreover, on an RTX3090, UE4-NeRF achieves higher frame rates at higher resolutions compared to Mega-NeRF.**
> > >
> > > ### [Comparison of whether Mega-NeRF enables dynamic rendering](https://github.com/JamChaos/UE4-NeRF/blob/master/pictures/mega_nerf_static_dyn_compare.jpg):
> > > In Mega-NeRF, enabling dynamic rendering results in a 2-second lag before the image becomes clear. While dynamic rendering does improve rendering quality to a certain extent, over time, Mega-NeRF's rendering frame rate drops to 15fps (and as low as 1fps during motion), leading to a suboptimal interactive rendering experience. **On an RTX3090, UE4-NeRF achieves a rendering speed of 50fps at a resolution of 2K, even at a higher rendering precision compared to Mega-NeRF. It also offers a seamless interactive rendering experience.**
> > >
> > > ### [Demonstration of Mega-NeRF's rendering results from far to near](https://github.com/JamChaos/UE4-NeRF/blob/master/pictures/mega_nerf_dyn_far_to_near.jpg):
> > > We presented images from Mega-NeRF at three distances showcasing dynamic rendering, ranging from far to near. It's evident that as the distance gets closer, Mega-NeRF does not enhance rendering precision further, indicating that it has reached its modeling precision limit. **In contrast, UE4-NeRF's multi-level mesh scene representation significantly improves rendering quality. When close to the surface, utilizing high-precision mesh representations.**
> > >
> > > Best regards and thank you once again.

---

> > > > ### Author Response · Authors · 2023-08-21
> > > > **Further Explanation of Visual Image Links.**
> > > >
> > > > **Dear Reviewer, we included links to images in each subheading. Kindly click on them to access the external images.**
> > > >
> > > > **To facilitate your review further, we  directly listed the links:**
> > > > - **Link 1.** https://github.com/JamChaos/UE4-NeRF/blob/master/pictures/ours_mega_nerf_compare.jpg
> > > > - **Link 2.**  https://github.com/JamChaos/UE4-NeRF/blob/master/pictures/mega_nerf_static_dyn_compare.jpg
> > > > - **Link 3.**  https://github.com/JamChaos/UE4-NeRF/blob/master/pictures/mega_nerf_dyn_far_to_near.jpg
> > > >
> > > >
> > > > **We are looking forward to receiving your response.**

---

> > > > > ### Comment · Reviewer_x4aP · 2023-08-21
> > > > >
> > > > > I appreciate the author's efforts in addressing my concerns regarding the comparison with Mega-NeRF methods. Given these efforts, I'm inclined to adjust my rating to 'borderline accept'. I anticipate that the author will integrate the aforementioned discussions and experimental insights into their forthcoming revisions.

---

> > > > > > ### Author Response · Authors · 2023-08-21
> > > > > >
> > > > > > Thank you for your response. We will add the complete experiment on the Mega-NeRF dataset and compare our method with Mega-NeRF as an additional experiment in the revision. Improving the rating gave us great encouragement!

---

### Author Rebuttal · Authors · 2023-08-10

## Comparison with Mobile-NeRF.
**Dataset** We test the performance of Mobile-NeRF in a block. It contains 239 pictures, each with a resolution of 6000x4000.

  In Table 1, we see that Mobile-NeRF takes 2 days to train just one block and requires 4x3090ti GPUs. If it trains the whole scene, it takes two months. And our method only need 40 minutes to train a block (1x3090ti), when training multiple blocks, we can perform multi-GPUs parallel training. In Figure 2, the left side is comparison of the rendered image under the training perspective, and the right side is comparison of the final result under free perspective in the renderer. Compared with Ground truth, UE4-NeRF shows amazing rendering results, and Mobile-NeRF produces obvious blurring. The results in the final renderer on the right show that our real-time rendering results are also much better than Mobile-NeRF. Mobile-NeRF finally uses webgl for real-time rendering. We speculate that the poor rendering ability of webgl has a certain impact on the rendering quality. WebGL sorts objects according to the coordinates of the center of the object instead of sorting according to the surface, and there will be various strange interlacing.
## Differences between Mobile-NeRF and UE4-NeRF.
1.The training strategy is different. Instead of Mobile-NeRF's three-stage training, we adjust weights of each part of the loss function in stages during training to control the tendency of the training process.

2.We train 5 levels of meshes at the same time, and the overhead of 5 levels of mesh training is only increased by 1/7 compared to training only the highest-precision mesh. And UE4-NeRF can control the rendering overhead by dynamically switching between different levels of precision of the model during rendering.

3.We use a different grid layout than Mobile-Nerf. In our mesh layout, the vertices converge faster.

4.The export strategy is different. Our method of calculating visibility is different from that of Mobile-NeRF, and we only save the structural relationship between vertices and faces and the UV coordinates of vertices, and do not directly save the features on the faces as texture maps.

5.The rendering strategy is different. Our approach is a dynamic loading and rendering strategy. The asynchronous process only takes less than 1.5 seconds. For comparison, Mobile-NeRF is an offline precomputation and real-time rendering strategy, which requires all the feature maps on all surfaces to be computed when the model is exported, resulting in exponentially higher hardware resource requirements in large scenes. We can infer higher-precision feature maps in the rendering stage without requiring additional storage space. We use higher precision 32x32 feature maps, while Mobile-NeRF uses 20x20 feature maps and requires more storage space.

6.The Mobile-NeRF is unable to model any translucent objects, but our method can approximate modeling and rendering translucent objects in UE4. We achieve our goal of rendering translucent objects by alpha dithering and temporal fusion.

7.Measuable.
## Comparison with Mega-NeRF.
**Dataset** We measure the performance of Mega-NeRF in a scene. The scene contains 2000 images, each with a resolution of 6000x4000.

For each block, Mega-NeRF needs 36 hours to train for 500,000 epochs, while we only need 40 minutes to train for 80,000 epochs to reach convergence. When training the entire scene, Mega-NeRF requires 12 hours of additional time, but UE4-NeRF only needs 1 hour. It is worth noting that Mega-Nerf generates a large number of temporary files during training. In Figure 1, The left side is comparison of the rendered image under the training perspective, and the right side is comparison of the final result under the free perspective in the renderer. Compared with Ground truth, UE4-NeRF shows amazing rendering results, and Mega-NeRF produces obvious blurring. The results in the final renderer on the right show that our real-time rendering results are also much better than Mega-NeRF. Real-time rendering results in Mega-nerf-viewer are weird. Through our experiments, the maximum resolution supported by Mega-nerf-viewer is only 800x800 (1x RTX3090ti), when we enlarge its resolution, the program crashes.
## Supplementary Explanation of NGP and Block Strategy.
The NGP approach employs dynamic resolution and multi-frame fusion to achieve high frame rates for real-time rendering. There is a noticeable presence of large pixel artifacts when the window size is larger. In NGP experiments, we assessed the rendering speed and computational cost at a fixed 2K rendering resolution. Even with a RTX 3090ti , it can only support rendering of a single high-complexity nerf model, yielding a frame rate of under 30fps. Further elevating the rendering resolution would lead to insufficient VRAM capacity, causing crash.

In the past, we also attempted to directly combine NGP with block to model large scenes. For a large scene, we use 32 lower-complexity NGP models to model each area, minimizing the memory usage of a single model's hash grid. When rendering, the method of dynamically loading and selecting models is combined to further reduce the occupation of GPU memory and computing resources.  The modeling accuracy was improved when using NGP with block strategy compared to using a single NGP model for the entire scene. However, despite these efforts, the rendering speed remained relatively low in practical. We could achieve approximately 10 fps at 720P resolution using two 3090Ti. Furthermore, doubling the resolution led to a twofold increase in VRAM usage and halved the frame rate. Therefore, only the method of combining the block strategy with NGP cannot realize high-resolution real-time rendering of the nerf model modeled in a large scene. Even if our method loads 32 blocks, it can achieve a rendering frame rate of 50fps at 2K resolution on a 3090ti, and can also achieve a relatively high rendering frame rate at 4K resolution.

---

### Author Response · Authors · 2023-08-20
**Dear Reviewers, we are anxiously waiting for your response.**

Dear Reviewers, thank you for raising your concerns. We provided comprehensive rebuttals addressing the issues you raised and included detailed supplementary experiments. ***With only one day left until the discussion concludes, we are anxiously waiting for your response. If you could respond to our rebuttal, we would greatly appreciate it. We are looking forward to further communication with you.***

**Thank you once again.**

---

### Decision · Program_Chairs · 2023-09-21

**Decision:**

Accept (poster)

**Comment:**

Reviewers and the AC read the rebuttal and took that into consideration for their final recommendation. The AC believes that the answers to the unresponsive reviewer in the rebuttal reasonably addressed their concerns. Many suggestions for improving the exposition, additional baseline comparisons, among other suggestions by reviewers should be included in the camera-ready paper.